# CC-Time: Cross-Model and Cross-Modality Time Series Forecasting

## Abstract

With the success of pre-trained language models (PLMs) in various application fields beyond natural language processing, language models have raised emerging attention in the field of time series forecasting (TSF) and have shown great prospects. However, current PLM-based TSF methods still fail to achieve satisfactory prediction accuracy matching the strong sequential modeling power of language models. To address this issue, we propose Cross-Model and Cross-Modality Learning with PLMs for time series forecasting (CC-Time). We explore the potential of PLMs for time series forecasting from two aspects: 1) what time series features could be modeled by PLMs, and 2) whether relying solely on PLMs is sufficient for building time series models. In the first aspect, CC-Time incorporates cross-modality learning to model temporal dependency and channel correlations in the language model from both time series sequences and their corresponding text descriptions. In the second aspect, CC-Time further proposes the cross-model fusion block to adaptively integrate knowledge from the PLMs and time series model to form a more comprehensive modeling of time series patterns. Extensive experiments on nine real-world datasets demonstrate that CC-Time achieves state-of-the-art prediction accuracy in both full-data training and few-shot learning situations.

**Resources:** https://anonymous.4open.science/r/CC-Time-7E86.

## 1 Introduction

With the rapid growth of the Internet of Things, vast amounts of time series data are being generated, driving increasing interest in time series forecasting (TSF) Kaastra & Boyd (1996); Faloutsos et al. (2018). Current TSF methods primarily design specific modules to exploit the inherent knowledge of the time series data, and achieve good prediction accuracy Liu et al. (2024c); Nie et al. (2023), which we call *time-series-specific models* in this paper.

Recently, pre-trained language models (PLMs) have demonstrated remarkable success across diverse fields Wang et al. (2024); Wu et al. (2024), prompting exploration in TSF Zhou et al. (2023); Jin et al. (2024a). Some approaches attempt to leverage the representation capacity and sequential modeling capability of PLMs to capture time series patterns for TSF, which we call *PLM-based models* Liu et al. (2024d). Although these methods show good prospects, they have not yet achieved satisfactory prediction accuracy, leaving an under-explored problem of how to effectively activate the potential of PLMs for TSF. Motivated by this, we raise and explore two important questions:

***What time series characteristics could be modeled by pre-trained LMs?*** Real-world multivariate time series exhibit two critical characteristics: (1) temporal dependencies across time steps and (2) correlations with different channels. Capturing these features is essential for modeling the underlying data structure and improving prediction performance. However, existing PLM-based methods mainly focus on modeling temporal dependency and typically adopt a channel-independent approach, overlooking the potential of leveraging text modality knowledge stored in PLMs to model channel correlations Zhou et al. (2023); Liu et al. (2024d). Meanwhile, time-series-specific methods are restricted to a single time-series modality, and they are more susceptible to numerical noise Cheng et al. (2023) and lack the capacity to model correlations from other perspectives, such as the semantic perspective. Consequently, effectively modeling temporal dependency and channel correlations with multi-modality knowledge is necessary.

*Is relying solely on pre-trained LMs sufficient for building time series models?* Recent studies indicate that PLM-based models and time-series-specific models focus on different aspects of modeling time series patternsJin et al. (2024a;b), and we also observe this phenomenon in the cross-model analysis of Section 4.4. Time-series-specific models excel at capturing basic patterns, such as trend and seasonality patterns Jin et al. (2024b), and language models possess strong semantic understanding capability and multi-domain knowledge, which provide additional analytical perspectives for forecasting Jin et al. (2024a); Liu et al. (2024d). Therefore, integrating these two types of models provides a more comprehensive understanding of time series patterns. However, a straightforward fusion, such as feature concatenation or knowledge distillation Phuong & Lampert (2019); Kim & Rush (2016), does not bridge the gap between semantic information and numerical representations. Furthermore, due to the heterogeneity of time-series-specific models and PLMs, the correspondence between knowledge from both models is unclear. Therefore, a novel fusion method to adaptively integrate knowledge from both models is necessary.

To address these challenges, we propose Cross-Model and Cross-Modality Modeling, namely **CC-Time**, to explore the potential of PLMs for TSF. CC-Time is a dual-branch framework that contains a PLM branch and a time series branch, together with their cross-model fusion.

**For the first aspect:** In the PLM branch, we propose cross-modality modeling with PLMs to capture temporal dependency and channel correlations, aiming at fully utilizing multi-modality knowledge in modeling complex time series patterns. In addition to capturing temporal dependency through patching with PLMs, we innovatively explore the potential of PLMs to model complex channel correlations by leveraging their stored knowledge. To enhance this process, we incorporate time series data with corresponding channel text descriptions as bimodal inputs, enabling PLMs to access both numerical patterns and semantic information for more complex and robust channel correlations. Importantly, these descriptions can be automatically acquired without requiring any additional human effort. **For the second aspect:** To better leverage the strength of both PLMs and time-series-specific models in capturing time series patterns, we propose a Cross-model Fusion Block (CMF Block) to adaptively integrate knowledge from the PLM branch and the time series branch of CC-Time. At each layer of CC-Time, the CMF Block leverages the current attention, memory attention, and gated fusion mechanism to adaptively fuse different-level features derived from the current layer and previous layers of the PLM branch. This fusion process makes the model capture complex features that encapsulate the semantic information and the intricate time series correlations. Subsequently, the CMF Block further integrates these features with features extracted from the time series branch. Overall, this adaptive cross-model fusion empowers CC-Time with a more comprehensive understanding of time series. Specifically, our contributions are as follows:

- We propose cross-modality modeling with PLMs to capture temporal dependency and channel correlations based on time series and corresponding text descriptions, which can be automatically acquired without requiring additional human effort, effectively mining and activating PLM knowledge related to time series.

- We further propose a cross-model fusion block to adaptively integrate knowledge from PLMs and time-series-specific models, empowering the model with a more comprehensive understanding of time series.

- Extensive experiments on nine datasets have demonstrated that CC-Time achieves state-of-the-art prediction accuracy in both full-data and few-shot situations.

## 2 RELATED WORK

**Channel Correlation Modeling** Channel correlation modeling has been proven to be essential for time series forecasting. Some existing methods adopt a channel-independent strategy Nie et al. (2023); Lin et al. (2024); Xu et al. (2024); Liu et al. (2024e), where the same weights are shared across all channels. Numerous studies employ Graph Neural Network (GNN) to capture the channel correlations Liu et al. (2022); Yi et al. (2023); Shang et al. (2021). MTGNN Wu et al. (2020) extends the application of GNN from spatio-temporal prediction to multivariate time series forecasting and proposes a method for computing an adaptive cross-channel graph. In addition to GNN-based methods, transformer-based models have made various attempts to capture correlations between channels Zhang & Yan (2023); Liu et al. (2024c); Yang et al. (2024); Wang et al. (2023). Crossformer Zhang & Yan (2023) proposed a two-stage attention layer to capture the cross-time and

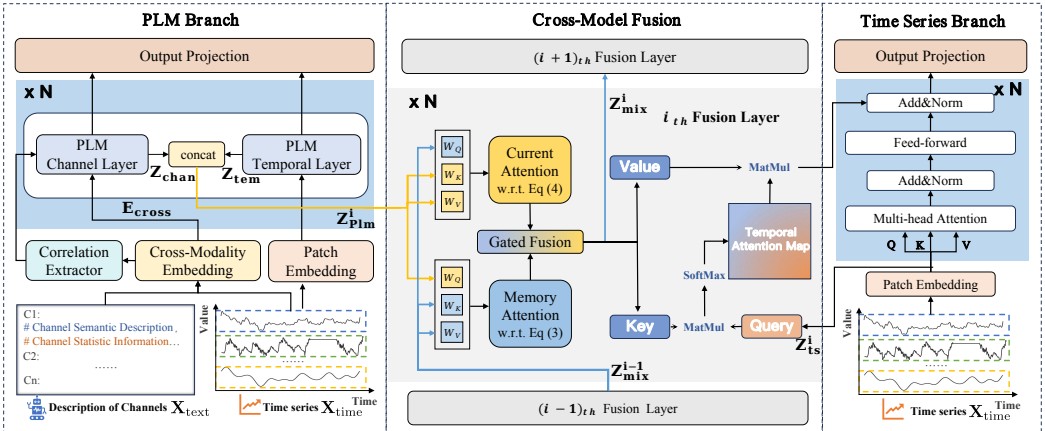

Figure 1: The framework of the proposed Cross-Model and Cross-Modality Modeling (CC-Time) consists of three components: the PLM Branch, Cross-Model Fusion, and the Time Series Branch.

cross-channel dependency efficiently. However, these models primarily design specific modules and only rely on time series modality to model correlations, which limits their capacity to fully capture complex channel correlations. CC-Time incorporates channel text descriptions and leverages language models to model channel correlations from a semantic perspective, empowering CC-Time's completeness of modeling correlations.

**Pre-trained LMs for Time Series Forecasting** Pre-trained language models (PLMs) make progress in various fields beyond natural language processing. Recently, numerous studies have utilized powerful sequence modeling and representation capabilities of PLMs to model complex time series patterns, showcasing their potential in forecasting Zhang et al. (2024); Jiang et al. (2024); Hu et al. (2025). These studies primarily involve direct usage Gruver et al. (2023); Xue & Salim (2024), parameter-efficient fine-tuning Zhou et al. (2023); Chang et al. (2023); Tan et al. (2024), prompting Liu et al. (2024d;b); Cao et al. (2024); Pan et al. (2024), and modal alignment Jin et al. (2024a); Sun et al. (2024); Liu et al. (2024a; 2025c;b;a). For example, GPT4TS Zhou et al. (2023) fine-tunes the limited parameters of PLMs, demonstrating competitive performance by transferring knowledge from large-scale pre-training text data. UniTime Liu et al. (2024b) designs domain instructions to align time series and text modality. Time-LLM Jin et al. (2024a) reprograms time series into text to align the representation of PLMs. However, existing PLM-based methods primarily use PLMs as simple feature extractors and have not fully exploited their potential for modeling time series patterns. CC-Time proposes cross-model fusion to combine the strengths of both language models and time-series-specific models, adaptively integrating their knowledge to achieve a more holistic understanding of time series.

## 3 METHODOLOGY

### 3.1 OVERALL ARCHITECTURE

To better exploit the potential of pre-trained language models (PLMs) for TSF and comprehensively model time series patterns, we propose cross-model and cross-modality modeling (**CC-Time**). As illustrated in Figure 1, it comprises three components: the PLM branch, the time series branch, and the cross-model fusion. In the PLM branch, we propose cross-modality modeling to fully utilize multi-modal knowledge in modeling temporal dependency and channel correlations. For temporal modeling, time series features are extracted by embedding time series patches and inputting them into the PLM temporal layer. For channel modeling, considering that directly modeling channel correlations solely from time series can easily be affected by numerical noise, we incorporate both channel text descriptions and time series as multi-modal inputs. By modeling channel correlations from the semantic space by the PLM channel layer, we obtain robust and complex correlation features. However, relying solely on PLMs is insufficient to fully model time series patterns. To address this, we further integrate the strengths of PLMs and time-series-specific models through a novel cross-

model fusion module. This module performs a two-step fusion: (1) adaptively fusing hierarchical features among layers within the PLM branch, and (2) performing cross-model fusion between the PLM and the time series branch. Through this novel fusion process, CC-Time effectively combines knowledge from both branches, enhancing its capability for time series understanding.

## 3.2 Cross-Modality Modeling with PLMs

In the PLM branch, we propose cross-modality modeling with PLMs to fully utilize multi-modality knowledge. Unlike existing PLM-based methods, we not only adopt a patching strategy to model temporal dependencies Zhou et al. (2023), but also, for the first time, exploit PLMs to capture complex channel correlations from a semantic perspective. Specifically, as shown in the PLM branch of Figure 1, we construct a *cross-modality embedding* by integrating channel text descriptions with time series, providing semantic context for channel modeling. Based on the embedding, we design *the PLM Channel Layer* that models sample-specific channel correlations from a semantic perspective. Together with the global correlations captured by *the Correlation Extractor* from the entire training data, they jointly form comprehensive channel representations. Finally, the channel representations are fused with temporal representations extracted by *the PLM Temporal Layer*, yielding the final output representation at each PLM layer.

**Cross-Modality Embedding** The time series modality introduces a perspective of building channel correlations based on the dynamic time series features. However, only focusing on this single modality limits the completeness of correlation modeling, as it does not well utilize language models' abilities to model complex semantic features. Motivated by this, we innovatively generate and utilize text descriptions of each channel for correlation modeling, which brings a new perspective on the semantic meanings of channels and naturally utilizes PLMs' powerful language processing and understanding capabilities.

Our proposed cross-modality embedding uses two modalities as the inputs: time series $\mathbf{X}_{\text{time}} \in \mathbb{R}^{C \times T}$ and text descriptions $\mathbf{X}_{\text{text}} \in \mathbb{R}^{C \times L}$, where $C$ represents the number of channels, and $T$ and $L$ represent the lengths of the time series and their text descriptions. For the input time series $\mathbf{X}_{\text{time}} \in \mathbb{R}^{C \times T}$, channel embedding is used to describe the overall temporal properties of each channel, with a linear mapping along the temporal dimension yielding $\mathbf{E}_{\text{chan}} \in \mathbb{R}^{C \times D_l}$, where $D_l$ represents the dimension of the embedded features. The corresponding text descriptions for each channel consist of two parts: channel semantic description and channel statistical information. The former provides a detailed semantic explanation for each channel, such as physical interpretations and causal relationships, and the latter offers relevant quantitative statistical details about these channels, such as the mean, variance. These descriptions about channels *can be automatically acquired without requiring any additional human effort*. The details about the channel text descriptions construction process are provided in Appendix A. To effectively integrate the embedding of the two modalities, we use linear mapping to compress the text embedding, aligning it with the dimension of channels. Then we add the text embedding and time series channels embedding to get the cross-modality embedding $\mathbf{E}_{\text{cross}} \in \mathbb{R}^{C \times D_l}$.

**Correlation modeling with PLMs** As illustrated in Figure 2, based on the cross-modality embedding as the input, the PLM channel layer leverages the PLMs' knowledge to model sample-specific channel correlations from a semantic perspective. In parallel, we introduce a correlation extractor to capture global correlations across the entire training data. These two modules complement each other, enabling more comprehensive and robust modeling of channel correlations. Given that channel correlations dynamically change over time, rely-

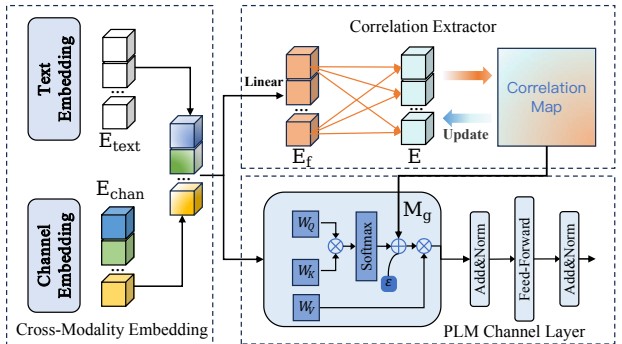

Figure 2: Cross-modality correlation modeling with PLMs.

ing solely on the current time series sample and corresponding text descriptions to capture sample-

specific correlations is not enough. To address this, the correlation extractor is designed to preserve global channel features across the entire training data while learning latent global correlations. Specifically, as shown in Figure 2, we initialize a learnable correlation extractor $\mathbf{E} \in \mathbb{R}^{C \times D_r}$ to preserve global channel features, where $D_r$ denotes the dimension of feature space. We use a linear projection to map the cross-modality embedding $\mathbf{E}_{\text{cross}}$ into the feature space of the correlation extractor $\mathbf{E}$ as current channel features $\mathbf{E}_{\text{f}}$. Subsequently, by utilizing the current channel features with the correlation extractor, a global correlation map $\mathbf{M}_{\text{g}} \in \mathbb{R}^{C \times C}$ is adaptively learned. The process is as follows:

$$\mathbf{M}_{\text{g}}^{i,j} = \frac{\exp(\mathbf{E}_{\text{f}}[i] * \mathbf{E}^T[j])}{\sum_{j=1}^{C} \exp(\mathbf{E}_{\text{f}}[i] * \mathbf{E}^T[j])}, \tag{1}$$

where $\mathbf{M}_{\text{g}}^{i,j}$ represents the weight of correlation between the $i$-th channel and the $j$-th channel. To make the correlations extracted from the extractor more global and generalizable, we use the current time series sample to update the correlation extractor. Specifically, we perform matrix multiplication between the global correlation map $\mathbf{M}_{\text{g}}$ and the extractor $\mathbf{E}$ to get new channel features, followed by weighted integration with the existing channel features to produce the updated extractor $\mathbf{E}'$.

To leverage pre-trained knowledge in the PLM, we reuse its layers to model correlations among channels, denoted as PLM channel layers. As illustrated in Figure 2, the cross-modality embedding $\mathbf{E}_{\text{cross}}$ is the input of the PLM Channel Layer, we perform linear transformations to obtain the query, key, and value in attention operations, denoted as $\mathbf{Q}_{\text{chan}}$, $\mathbf{K}_{\text{chan}}$, and $\mathbf{V}_{\text{chan}} \in \mathbb{R}^{D_l \times D_l}$. Then the current attention map $\mathbf{M}_{\text{c}} \in \mathbb{R}^{C \times C}$ is computed from the query and key, which serves as the correlation map of the current time series. To capture complex channel correlations and mitigate the focus on only the current time series, we perform a weighted fusion operation on the current correlation map and the global correlation map from the correlation extractor to obtain the final correlation map. Then we perform matrix multiplication with $\mathbf{V}_{\text{chan}}$ to obtain the output representation $\mathbf{Attn}_{\text{chan}} \in \mathbb{R}^{C \times D_l}$ of the attention process:

$$\mathbf{M}_{\text{c}} = \text{Softmax}(\mathbf{Q}_{\text{chan}}\mathbf{K}_{\text{chan}}^T / \sqrt{D_l}), \quad \mathbf{Attn}_{\text{chan}} = (\epsilon\mathbf{M}_{\text{g}} + (1 - \epsilon)\mathbf{M}_{\text{c}})\mathbf{V}_{\text{chan}} \tag{2}$$

where $\epsilon$ is the parameter to balance the global correlation and the current local correlation. After attention operation, a feed-forward network is used to process the representation of each channel to obtain the final output $\mathbf{Z}_{\text{chan}} \in \mathbb{R}^{C \times D_l}$ of the PLM channel layer.

**Temporal modeling with PLMs** For temporal modeling, considering the sequential modeling capability of PLMs, we simultaneously leverage the pre-trained LM layers to capture complex temporal patterns, denoted as PLM temporal layers. Specifically, we adopt a patching strategy with PLMs Zhou et al. (2023), where the time series $\mathbf{X}_{\text{time}} \in \mathbb{R}^{C \times T}$ is divided into $N_p$ patches based on a patch size $S$. These patches are then passed through patch embedding to obtain $\mathbf{E}_{\text{patch}} \in \mathbb{R}^{C \times N_p \times D_l}$, which is subsequently fed into the PLM temporal layer to model temporal dependencies, resulting in the final output $\mathbf{Z}_{\text{tem}} \in \mathbb{R}^{C \times N_p \times D_l}$.

To integrate language models' knowledge from both temporal and correlation modeling, we perform concatenation fusion on the temporal features $\mathbf{Z}_{\text{tem}}$ and the correlation features $\mathbf{Z}_{\text{chan}}$ to get the complex features $\mathbf{Z}_{\text{plm}} \in \mathbb{R}^{C \times N_m \times D_l}$, where $N_m$ denotes the number of concatenated patches.

### 3.3 CROSS-MODEL FUSION

Considering the advantage of PLMs and time-series-specific models, specific models focus on modeling patterns from numerical representations, while PLMs demonstrate strong generalization and complex-patterns-modeling capabilities, which provide additional perspectives for forecasting. Motivated by this, we propose the cross-model fusion to adaptively integrate knowledge from the two types of models, thus fully leveraging their respective strengths. Our proposed fusion module performs a two-step fusion: 1) adaptively fusing hierarchical features among layers from the PLMs to enhance the richness of the PLM representations, and 2) cross-model fusion between the PLM branch and the time series branch to construct a comprehensive understanding of time series.

**Time series Branch modeling** We use the time series as the input to model time series patterns. Similar to the patch embedding in the PLM branch, the time series $\mathbf{X}_{\text{time}} \in \mathbb{R}^{C \times T}$ is divided into $N_p$ patches and then processed by patch embedding to get $\mathbf{E}'_{\text{patch}} \in \mathbb{R}^{C \times N_p \times D_t}$, where $D_t$ is the

feature dimension of the time series model. As depicted in Figure 1, the time series branch is based on a transformer structure. For the $i_{th}$ layer, the transformer layer extracts the temporal dynamics between patches and obtains the output $\mathbf{Z}_{\text{ts}}^i \in \mathrm{R}^{C \times N_p \times D_t}$ of the layer as time series features.

**Cross-model fusion Block**  Due to the heterogeneity of time series models and language models, the correspondence between knowledge from these two types of models is unclear. Direct layer-to-layer integration of features captured by these models could lead to feature mismatch. Therefore, we first adaptively fuse the PLM features derived from the current layer and the previous layers to get comprehensive complex features. Specifically, for the adaptive fusion process at the $i_{th}$ layer, the Cross-model Fusion Block (CMF block) takes the mixed features from the $i - 1_{th}$ layer, denoted as $\mathbf{Z}_{\text{mix}}^{i-1} \in \mathbb{R}^{C \times N_m \times D_l}$, and the features $\mathbf{Z}_{\text{plm}}^i \in \mathbb{R}^{C \times N_m \times D_l}$ of current layer. To better integrate these features, we propose two types of attention mechanisms, called *memory attention* and *current attention*. Memory attention uses the features $\mathbf{Z}_{\text{plm}}^i$ from the current layer as a query to selectively focus on the accumulated features from the previous layer, yielding $\mathbf{Attn}_{\text{mix}} \in \mathbb{R}^{C \times N_m \times D_l}$.

$$\mathbf{Q}_{\text{plm}} = \mathbf{Z}_{\text{plm}}^i \mathbf{W}_{Q_c}, \mathbf{K}_{\text{mix}} = \mathbf{Z}_{\text{mix}}^{i-1} \mathbf{W}_{K_m}, \mathbf{V}_{\text{mix}} = \mathbf{Z}_{\text{mix}}^{i-1} \mathbf{W}_{V_m},$$
$$\mathbf{Attn}_{\text{mix}} = \text{Softmax}(\mathbf{Q}_{\text{plm}} \mathbf{K}_{\text{mix}}^T / \sqrt{D_l}) \mathbf{V}_{\text{mix}}. \tag{3}$$

In contrast, current attention uses the output complex features $\mathbf{Z}_{\text{mix}}^{i-1}$ from the previous layer as a query to dynamically focus on the features from the current layer, resulting in $\mathbf{Attn}_{\text{plm}} \in \mathbb{R}^{C \times N_m \times D_l}$.

$$\mathbf{Q}_{\text{mix}} = \mathbf{Z}_{\text{mix}}^{i-1} \mathbf{W}_{Q_m}, \mathbf{K}_{\text{plm}} = \mathbf{Z}_{\text{plm}}^i \mathbf{W}_{K_c}, \mathbf{V}_{\text{plm}} = \mathbf{Z}_{\text{plm}}^i \mathbf{W}_{V_c},$$
$$\mathbf{Attn}_{\text{plm}} = \text{Softmax}(\mathbf{Q}_{\text{mix}} \mathbf{K}_{\text{plm}}^T / \sqrt{D_l}) \mathbf{V}_{\text{plm}}. \tag{4}$$

Then we leverage gated fusion to adaptively fuse $\mathbf{Attn}_{\text{plm}}$ and $\mathbf{Attn}_{\text{mix}}$ to get the mixed features $\mathbf{Z}_{\text{mix}}^i$ for $i_{th}$ layer:

$$\mathbf{Z}_{\text{mix}}^i = \beta \mathbf{Attn}_{\text{mix}} + (1 - \beta) \mathbf{Attn}_{\text{plm}}, \tag{5}$$

where $\beta$ is a learnable parameter, controlling the fusion of current features and accumulated features.

Based on the mixed cross-layer features $\mathbf{Z}_{\text{mix}}^i$ from the PLM branch and the features $\mathbf{Z}_{\text{ts}}^{i-1}$ from the time series branch, we further perform cross-model fusion between these two models. Specially, we perform learnable linear transformations on $\mathbf{Z}_{\text{mix}}^i$ to get the key and value, denoted as $\mathbf{K}_{\text{cross}}$ and $\mathbf{V}_{\text{cross}} \in \mathrm{R}^{C \times N_m \times D_t}$, and use $\mathbf{Z}_l^{i-1}$ to obtain query $\mathbf{Q}_{\text{cross}} \in \mathrm{R}^{C \times N_p \times D_t}$. Then we compute the cross attention to get the fused features $\mathbf{Attn}_{\text{cross}} \in \mathrm{R}^{C \times N_p \times D_t}$ corresponding to the time series:

$$\mathbf{Attn}_{\text{corss}} = \text{Softmax}(\mathbf{Q}_{\text{cross}} \mathbf{K}_{\text{cross}}^T / \sqrt{D_t}) \mathbf{V}_{\text{cross}}. \tag{6}$$

Finally, the features $\mathbf{Attn}_{\text{cross}}$ and the features $\mathbf{Z}_{ts}^i$ from the time series branch are added to obtain the cross-model features $\mathbf{Z}_{\text{cross}}^i \in \mathrm{R}^{C \times N_p \times D_t}$ for $i_{th}$ layer.

## 3.4 TRAIN AND INFERENCE

During the model training phase, to enhance training effectiveness and maintain the advantages of pre-trained language models, we freeze most of the parameters in the PLM branch and only fine-tune the positional encoding and layer normalization. All parameters in the cross-model fusion Block and time series branch are fine-tuned. For the training loss function, we calculate the loss between the outputs of the PLM branch and time series branch, $\hat{\mathbf{Y}}_{\text{plm}}$ and $\hat{\mathbf{Y}}_{\text{ts}}$, and the ground truth $\mathbf{Y}$, the total loss is as follows:

$$\mathcal{L}_{total} = \lambda |\hat{\mathbf{Y}}_{\text{plm}} - \mathbf{Y}| + (1 - \lambda)|\hat{\mathbf{Y}}_{\text{ts}} - \mathbf{Y}|, \tag{7}$$

where $\lambda$ is the hyperparameter. In the inference stage, only the output from the time series branch $\hat{\mathbf{Y}}_{\text{ts}}$ is used as the model prediction in the inference stage.

## 4 EXPERIMENT

### 4.1 EXPERIMENTAL SETUP

**Datasets**  To evaluate the prediction accuracy of CC-Time, we select nine real-world time series benchmarks from various domains, including energy, weather, nature, and traffic. These datasets include ETT (ETTh1, ETTh2, ETTm1, ETTm2), Weather, Electricity, Traffic, ZafNoo, and CzeLan.

**Baselines**   We select eight state-of-the-art models for time series forecasting as baselines, including PLM-based models: S$^2$IP-LLM Pan et al. (2024), FSCAHu et al. (2025), Time-LLM Jin et al. (2024a), UniTime Liu et al. (2024b), and GPT4TS Zhou et al. (2023), and time-series-specific models: iTransformer Liu et al. (2024c), Crossformer Zhang & Yan (2023), PatchTST Nie et al. (2023).

**Settings**   For a fair comparison, we set the input length $T$ to 96 and the output length $F$ to 96, 192, 336, and 720 for all baseline models and CC-Time. At the same time, all models do not use the drop last strategy Qiu et al. (2024). Refer to GPT4TS Zhou et al. (2023), we set GPT2 as the default architecture for the PLM branch of CC-Time and all pre-trained LM-based baselines. Meanwhile, in the PLM Layers of CC-Time, we only fine-tune the positional encoding and layer normalization to reduce learnable parameters.

## 4.2 Main Results

**Full-data Forecasting**   As illustrated in Table 1, CC-Time achieves state-of-the-art prediction accuracy, demonstrating the effectiveness of the model. Specifically, neither the PLM-based methods nor the time-series-specific methods consistently achieve the second-best accuracy, indicating that relying solely on PLMs or specific models is suboptimal. Therefore, an effective cross-model modeling approach proves to be promising. Compared to time-series-specific models, CC-Time outperforms the best model by 7.8% and 8.1% in MSE and MAE metrics. Compared to PLM-based models, CC-Time outperforms the best baseline by 7.9% and 8.9% in MSE and MAE metrics. PLM-based methods perform well on smaller datasets like ETTh1 and ETTh2, indicating that these models exhibit strong generalization capabilities and are well-suited for scenarios with limited or sparse data. For datasets with strong channel correlations, such as Electricity and Traffic, CC-Time consistently outperforms correlation modeling methods like iTransformer and Crossformer. We also compare CC-Time with recent time series foundation models in the Appendix D.

| Model | CC-Time | | FSCA | | S$^2$IP-LLM | | Time-LLM | | UniTime | | GPT4TS | | PatchTST | | iTransformer | | Crossformer | |
|---|---|---|---|---|---|---|---|---|---|---|---|---|---|---|---|---|---|---|
| Metric | MSE | MAE | MSE | MAE | MSE | MAE | MSE | MAE | MSE | MAE | MSE | MAE | MSE | MAE | MSE | MAE | MSE | MAE |
| ETTm1 | **0.375** | **0.377** | 0.398 | 0.406 | 0.396 | 0.395 | 0.410 | 0.409 | 0.385 | 0.399 | 0.389 | 0.397 | 0.387 | 0.400 | 0.407 | 0.409 | 0.513 | 0.495 |
| ETTm2 | **0.274** | **0.314** | 0.281 | 0.324 | 0.282 | 0.327 | 0.296 | 0.340 | 0.293 | 0.334 | 0.285 | 0.331 | 0.280 | 0.326 | 0.288 | 0.332 | 0.757 | 0.610 |
| ETTh1 | **0.424** | **0.424** | 0.430 | 0.437 | 0.447 | 0.439 | 0.446 | 0.443 | 0.442 | 0.447 | 0.447 | 0.436 | 0.468 | 0.454 | 0.454 | 0.447 | 0.529 | 0.522 |
| ETTh2 | **0.363** | **0.389** | 0.374 | 0.402 | 0.384 | 0.408 | 0.389 | 0.408 | 0.377 | 0.402 | 0.381 | 0.408 | 0.386 | 0.406 | 0.383 | 0.406 | 0.942 | 0.683 |
| Weather | **0.240** | **0.260** | 0.255 | 0.274 | 0.261 | 0.281 | 0.274 | 0.290 | 0.253 | 0.276 | 0.264 | 0.284 | 0.258 | 0.280 | 0.257 | 0.277 | 0.258 | 0.315 |
| Electricity | **0.174** | **0.257** | 0.187 | 0.237 | 0.198 | 0.283 | 0.223 | 0.309 | 0.215 | 0.304 | 0.205 | 0.290 | 0.204 | 0.290 | 0.178 | 0.269 | 0.244 | 0.344 |
| Traffic | **0.427** | **0.262** | 0.466 | 0.300 | 0.486 | 0.315 | 0.541 | 0.358 | 0.480 | 0.308 | 0.488 | 0.317 | 0.481 | 0.304 | 0.428 | 0.282 | 0.549 | 0.304 |
| ZafNoo | **0.545** | **0.434** | 0.589 | 0.473 | 0.577 | 0.470 | 0.591 | 0.481 | 0.581 | 0.478 | 0.594 | 0.477 | 0.576 | 0.466 | 0.577 | 0.471 | 0.550 | 0.449 |
| CzeLan | **0.240** | **0.261** | 0.273 | 0.299 | 0.262 | 0.290 | 0.272 | 0.303 | 0.275 | 0.306 | 0.273 | 0.295 | 0.268 | 0.298 | 0.274 | 0.302 | 0.914 | 0.585 |

Table 1: Time series forecasting results with the input length $T = 96$ and the prediction length $F = \{96, 192, 336, 720\}$. Bold: the best and underline: the second best. Complete results are in Table 12, and comparisons with time series foundation models are provided in Appendix D.

**Few-shot Forecasting**   We conduct few-shot forecasting using 10% of the training data on the ETT and Weather datasets to assess the few-shot learning capabilities of PLMs. As shown in Table 2, our proposed CC-Time achieves state-of-the-art prediction accuracy. Overall, CC-Time and PLM-based methods significantly outperform time-series-specific methods like PatchTST and iTransformer. This indicates that PLM-based methods have strong generalization capabilities, making them well-suited for scenarios with limited or sparse time series data, highlighting the potential of PLMs for time series forecasting.

## 4.3 Ablation Studies

**Cross-model Fusion**   We conduct substitution experiments using four methods: feature summation, feature concatenation, attention fusion, and knowledge distillation with five different random seeds. We also conduct ablation experiments on each specific module of the cross-model fusion in Appendix E. To further evaluate the fusion effect, we also compare their performance against using only the PLM branch. As shown in Figure 3 (a), compared to these existing fusion methods, our proposed fusion method achieves significant results, demonstrating the effectiveness of our cross-model

| Method | CC-Time | | FSCA | | S²IP-LLM | | Time-LLM | | UniTime | | GPT4TS | | PatchTST | | iTransformer | | Crossformer | |
|---|---|---|---|---|---|---|---|---|---|---|---|---|---|---|---|---|---|---|
| Metric | MSE | MAE | MSE | MAE | MSE | MAE | MSE | MAE | MSE | MAE | MSE | MAE | MSE | MAE | MSE | MAE | MSE | MAE |
| ETTm1 | **0.406** | **0.399** | 0.422 | 0.416 | 0.429 | 0.421 | 0.424 | 0.413 | 0.424 | 0.419 | 0.415 | 0.408 | 0.421 | 0.416 | 0.450 | 0.431 | 0.635 | 0.590 |
| ETTm2 | **0.291** | **0.335** | 0.308 | 0.346 | 0.310 | 0.347 | 0.306 | 0.347 | 0.303 | 0.346 | 0.300 | 0.345 | 0.300 | 0.347 | 0.305 | 0.349 | 1.226 | 0.805 |
| ETTh1 | **0.459** | **0.448** | 0.477 | 0.457 | 0.484 | 0.461 | 0.479 | 0.462 | 0.482 | 0.459 | 0.470 | 0.457 | 0.479 | 0.458 | 0.660 | 0.551 | 0.834 | 0.689 |
| ETTh2 | 0.412 | **0.416** | 0.416 | 0.423 | 0.443 | 0.443 | **0.410** | 0.420 | 0.425 | 0.431 | 0.419 | 0.425 | 0.423 | 0.424 | 0.435 | 0.439 | 1.225 | 0.845 |
| Weather | **0.259** | **0.275** | 0.268 | 0.289 | 0.265 | 0.288 | 0.273 | 0.290 | 0.270 | 0.289 | 0.270 | 0.288 | 0.271 | 0.285 | 0.272 | 0.290 | 0.593 | 0.592 |

Table 2: 10% few shot forecasting results with the input length T = 96 and the prediction length F = {96, 192, 336, 720}. Bold: the best and underline: the second best. Full results are in Table 13.

fusion. The attention fusion method performs second best, indicating that it partially integrates the corresponding knowledge. However, it lacks an adaptive process, leading to incomplete knowledge utilization from these two models.

**Cross-modality Modeling** We perform ablation studies on the text description, correlation extractor, and cross-modality correlation learning with five random seeds. Figure 3 (b) shows the distinct impact of each module. Removing cross-modality correlation learning significantly decreased prediction accuracy, particularly on the CzeLan dataset, indicating that cross-modality learning effectively and comprehensively models complex channel correlations. The text description provides semantic information to PLMs, aiding them in understanding complex channel correlations from different perspectives, thereby improving prediction accuracy. The correlation extractor learns global correlations from the dataset and complements the sample-specific correlations extracted by the PLM channel layer, leading to more comprehensive modeling of correlations.

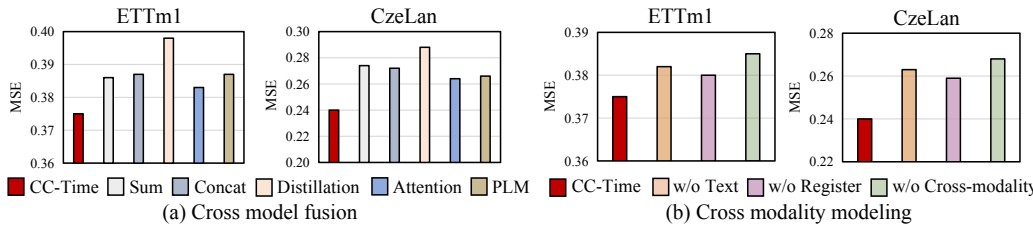

Figure 3: Ablation about cross-model fusion and cross-modality modeling with five different random seeds, with observed variations bounded within ±0.0015.

### 4.4 MODEL ANALYSIS

Due to the limitation of space, we conduct the channel text quality analysis, time series branch analysis, sensitivity analysis, efficiency analysis, and varying the input length in Appendix F.3, F.4, F.2, F.6, and F.7.

**Replacement of PLM Architecture** We replace GPT-2 with several more advanced pre-trained language models, including Flan-T5-250M Chung et al. (2024), LLaMA-7B Touvron et al. (2023), and LLaMA-13B. As shown in Table 3, different PLM architectures exhibit varying impacts on prediction performance. Overall, adopting stronger PLMs leads to improved forecasting accuracy in CC-Time, demonstrating its ability to leverage the knowledge of PLMs. Interestingly, we observe that the performance improvement from LLaMA-7B to LLaMA-13B is relatively marginal. This suggests that CC-Time may not heavily rely on the additional capabilities of larger models (e.g., reasoning), but instead primarily benefits from their semantic understanding and representation learning. Further experiments about the effectiveness of PLMs are in Appendix F.1.

**Cross-modality Correlation Modeling Analysis** To evaluate the effectiveness of correlation modeling, we select two baselines: iTransformer and GPT4TS, to compare with CC-Time. Inspired by the experiment demonstrations of the channel correlations in iTransformer, we calculate the Pearson Correlation Coefficients, a commonly used metric for approximately assessing correlations, between the predicted series and the future series for each model and normalize these results to

| Models | GPT2 | | Flan-T5-250m | | Llama-7B | | Llama-13B | |
|---|---|---|---|---|---|---|---|---|
| Metrics | MSE | MAE | MSE | MAE | MSE | MAE | MSE | MAE |
| ETTh1 | 0.424 | 0.424 | 0.435 | 0.442 | 0.420 | 0.418 | 0.419 | 0.418 |
| ETTh2 | 0.363 | 0.389 | 0.368 | 0.396 | 0.356 | 0.380 | 0.355 | 0.381 |

Table 3: Performance of CC-Time with different modality knowledge of PLMs.

analyze correlation modeling. We provide a case visualization in Figure 4. Compared to GPT4TS, a channel-independent method, iTransformer and CC-Time effectively model correlations. Compared to iTransformer, the channel correlations predicted by CC-Time are closer to the correlations from the future series, indicating that the correlation modeling of CC-Time is more effective and comprehensive. These observations also further illustrate the potential of PLMs for correlation modeling.

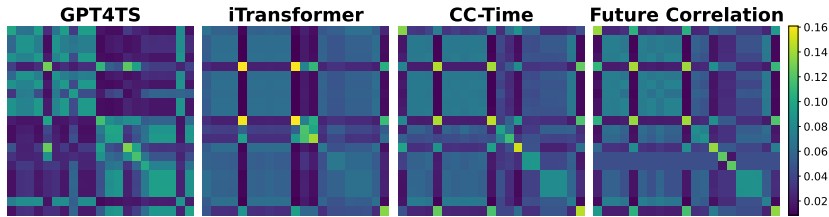

Figure 4: Visualization of channel correlations for different time series forecasting models.

**Cross-model Analysis** We analyze the modeling approaches of CC-Time, PLM-based methods, and time-series-specific methods. We use a standard metric: the centered kernel alignment (CKA) Kornblith et al. (2019) to assess the similarity between features and original data. A higher CKA indicates greater similarity, suggesting that the model learns simpler features, while a lower CKA suggests the model learns more complex features. As shown in Figure 5, time-series-specific methods exhibit high CKA values, while PLM-based methods exhibit low CKA values. This suggests that the features learned by the two types of methods differ significantly. The CKA similarity value of CC-Time is intermediate between the two categories of methods, and it achieves the lowest MSE, indicating that CC-Time effectively integrates characteristics from both types of models. Furthermore, CC-Time achieves the best prediction accuracy on two different types of datasets, demonstrating that this cross-model modeling approach can better adapt to diverse time series.

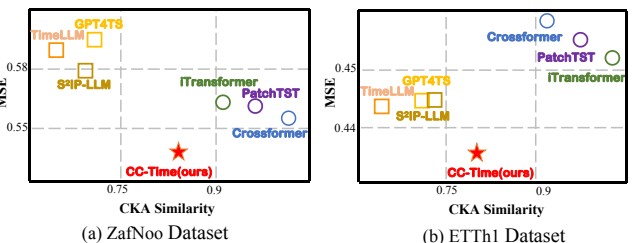

Figure 5: Cross-model analysis. Higher CKA values indicate that the model captures simpler features.

## 5 CONCLUSION

In this paper, we propose Cross-Model and Cross-Modality time series forecasting, namely CC-Time, to comprehensively model time series patterns. To leverage the capability of LLM for modeling complex patterns, CC-Time incorporates cross-modality modeling to capture temporal dependency and channel correlations in the LLMs from both time series sequences and their corresponding channel text descriptions. Furthermore, CC-Time proposes the cross-model fusion block to adaptively integrate knowledge from the LLMs and time-series-specific models to form a more comprehensive modeling of time series patterns. These innovative designs empower CC-Time to achieve state-of-the-art prediction accuracy in both full-data training and few-shot learning situations. At the same time, CC-Time exhibits better efficiency compared with most LLM-based methods. We provide our code at https://anonymous.4open.science/r/CC-Time-7E86.

ETHICS STATEMENT

Our work is conducted on publicly available benchmark datasets, without involving any additional personal information. No human subjects are involved in this research.

REPRODUCIBILITY STATEMENT

The performance of CC-Time and the datasets used in our work are real, and all experimental results can be reproduced. We have released the code of CC-Time in an anonymous repository: https://anonymous.4open.science/r/CC-Time-7E86.

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

# A  CHANNEL DESCRIPTIONS

As shown in Figure 6, for generating semantic descriptions, we construct prompts for each channel: *This is <dataset name> from <domain>, including <channel name1>, <channel name2>, ..., please describe these channels and their correlations.* We just need to input the dataset name, domain name, and the specific channel names into the predefined prompt. Then, by entering it into a large language model like ChatGPT, the semantic descriptions of channels are generated. For example, in the Weather dataset Wu et al. (2021), the description of the temperature channel (denoted as "T (degC)" in the dataset) is generated from LLMs as follows: *Temperature is a key parameter that describes climate conditions. It is closely related to humidity. As the temperature rises, the air can hold more moisture, affecting humidity.* CC-Time can leverage this semantic information to understand the correlation between temperature and humidity in the real world, which helps in better modeling patterns.

For statistic information, we calculate the maximum value, minimum value, mean, variance, and other statistics for each time series channel. Then, the semantic description and statistical information for each channel are concatenated to form the complete text description of each channel. These text descriptions $\mathbf{X}_{\text{text}}$ are input into a pre-trained text tokenizer to obtain text embedding results $\mathbf{E}_{\text{text}} \in \mathbb{R}^{C \times L \times D_l}$.

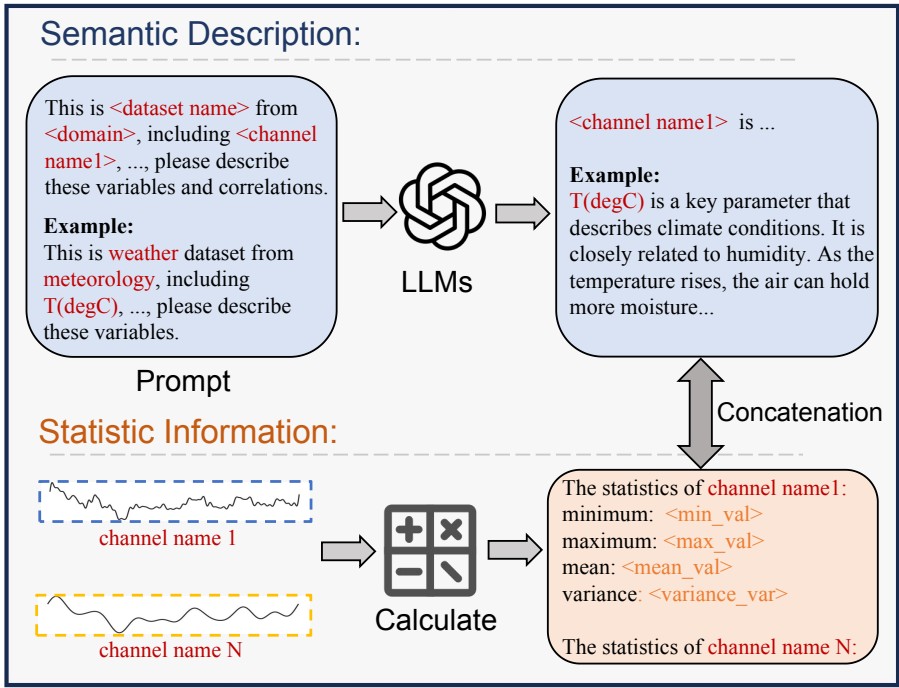

Figure 6: The construction process of channel text descriptions $\mathbf{X}_{\text{text}}$.

# B  DATASETS

ETT (Electric Transformer Temperature) Zhou et al. (2021) collected from two different electric transformers, spans from July 2016 to July 2018, and includes 7 channels. ETT is divided into four subsets: ETTh1 and ETTh2, recorded hourly, and ETTm1 and ETTm2, recorded every 15 minutes. (2) Weather Wu et al. (2021) includes 21 different meteorological indicators that provide comprehensive data on weather conditions. These indicators, such as temperature, barometric pressure, humidity and others, offer a broad perspective on the atmospheric environment. (3) Electricity Trindade (2015) contains the electricity consumption of 321 customers from July 2016 to July 2019, recorded hourly. (4) Traffic Wu et al. (2021) contains road occupancy rates measured by 862 sensors on freeways in the San Francisco Bay Area from 2015 to 2016, recorded hourly. (5) ZafNoo Poyatos et al. (2020) is from the Sapflux data project including sap flow measurements and environmental

variables. (6) CzeLan Poyatos et al. (2020) is from the Sapflux data project including sap flow measurements and environmental variables. We split each evaluation dataset into train-validation-test sets and detailed descriptions of evaluation datasets are shown in Table 4. According to the channel description generation method proposed in section 3.2, the corresponding text for datasets is generated without requiring any additional human effort.

| Dataset | Domain | Frequency | Lengths | Channels | Split |
|---------|--------|-----------|---------|----------|-------|
| ETTh1 | Electricity | 1 hour | 14400 | 7 | 6:2:2 |
| ETTh2 | Electricity | 1 hour | 14400 | 7 | 6:2:2 |
| ETTm1 | Electricity | 15 mins | 57600 | 7 | 6:2:2 |
| ETTm2 | Electricity | 15 mins | 57600 | 7 | 6:2:2 |
| Electricity | Electricity | 1 hour | 26304 | 321 | 7:1:2 |
| Weather | Environment | 10 mins | 52696 | 21 | 7:1:2 |
| Traffic | Transport | 1 hour | 17544 | 862 | 7:1:2 |
| ZafNoo | Nature | 30 mins | 19225 | 11 | 7:1:2 |
| CzeLan | Nature | 30 mins | 19934 | 11 | 7:1:2 |

Table 4: Detailed dataset descriptions.

## C  BASELINES

- S$^2$IP-LLM Pan et al. (2024) aligns the semantic space of pre-trained PLMs with time series embedding space and enhances the representation of time series using semantic space informed prompting to improve forecasting performance.

- FSCA Hu et al. (2025) builds the consistent context through structural alignment, logical alignment, and dual-scale GNNs, enabling PLMs to better understand time series.

- Time-LLM Jin et al. (2024a) aligns the time series features into the language feature space through reprogramming techniques, then concatenates them with the text prompt before inputting them into a pre-trained large language model for feature extraction.

- UniTime Liu et al. (2024b) designs domain instructions as prompts and uses the language model architecture for multi-source time series pre-training to extract broad knowledge.

- GPT4TS Zhou et al. (2023) uses the pre-trained GPT2 as the backbone and adapts it to the time series space for feature extraction by fine-tuning the positional encoding and normalization layers.

- iTransformer Liu et al. (2024c) treats the entire set of variables as tokens and uses the Transformer architecture to model the correlations between the entire set of channels.

- Crossformer Zhang & Yan (2023) proposes a two-stage attention mechanism (cross-time attention and cross-dimension attention) to model the dynamics of time and the correlations between channels.

- PatchTST Nie et al. (2023) employs a patching strategy to preserve local information and uses the Transformer architecture to capture the correlation between different patches for temporal modeling. It also applies a channel-independent strategy on the channel dimension.

- FEDformer Zhou et al. (2022) proposes a frequency-enhanced decomposed Transformer architecture to model temporal dynamics from the perspective of frequency.

- TimesNet Wu et al. (2023) transforms one-dimensional time series into a two-dimensional structure using Fourier transforms, enabling the modeling of 2D temporal variations to capture multi-periodicity time series patterns.

## D    COMPARED WITH TIME SERIES FOUNDATION MODELS

To further validate the prediction performance of CC-Time, we conduct comparative experiments with several state-of-the-art time series foundation models, including Chronos Ansari et al. (2024), MORIAI Woo et al. (2024), Timer Liu et al. (2024e), and TTMs Ekambaram et al. (2024). The evaluation is performed on ETT and Weather datasets, with input lengths of $\{96, 336, 512\}$ and output lengths spanning $\{96, 192, 336, 720\}$. The final results are selected based on the optimal performance across the three input lengths. As shown in Table 5, CC-Time achieves state-of-the-art prediction performance in most forecasting scenarios, outperforming the four baseline models. This demonstrates that CC-Time not only exhibits strong generalization capabilities but also effectively leverages knowledge from pre-trained language models, leading to significant improvements in time series forecasting accuracy.

| Models | | CC-Time | | Chronos | | MORIAI | | Timer | | TTMs | |
|---|---|---|---|---|---|---|---|---|---|---|---|
| Metrics | | MSE | MAE | MSE | MAE | MSE | MAE | MSE | MAE | MSE | MAE |
| ETTh1 | 96 | **0.357** | 0.389 | 0.388 | **0.387** | 0.394 | 0.399 | 0.413 | 0.424 | 0.361 | 0.392 |
| | 192 | 0.397 | **0.415** | 0.440 | 0.416 | 0.430 | 0.422 | 0.487 | 0.459 | **0.393** | 0.415 |
| | 336 | 0.415 | **0.429** | 0.477 | 0.434 | 0.450 | 0.437 | 0.501 | 0.471 | **0.411** | **0.429** |
| | 720 | 0.436 | 0.453 | 0.474 | 0.446 | 0.457 | 0.458 | 0.538 | 0.505 | **0.426** | **0.450** |
| ETTh2 | 96 | **0.265** | **0.328** | 0.292 | **0.328** | 0.285 | 0.329 | 0.324 | 0.366 | 0.270 | 0.330 |
| | 192 | **0.342** | 0.376 | 0.362 | **0.371** | 0.352 | 0.374 | 0.409 | 0.410 | 0.362 | 0.384 |
| | 336 | **0.359** | **0.395** | 0.400 | 0.404 | 0.384 | 0.403 | 0.419 | 0.428 | 0.367 | 0.400 |
| | 720 | **0.382** | 0.421 | 0.412 | **0.420** | 0.419 | 0.432 | 0.451 | 0.456 | 0.384 | 0.425 |
| ETTm1 | 96 | **0.276** | **0.327** | 0.339 | 0.340 | 0.464 | 0.404 | 0.335 | 0.359 | 0.285 | 0.336 |
| | 192 | **0.323** | **0.354** | 0.392 | 0.372 | 0.488 | 0.422 | 0.424 | 0.406 | 0.326 | 0.363 |
| | 336 | **0.354** | **0.374** | 0.440 | 0.398 | 0.520 | 0.442 | 0.450 | 0.428 | 0.357 | 0.380 |
| | 720 | **0.412** | **0.407** | 0.530 | 0.442 | 0.598 | 0.482 | 0.514 | 0.465 | 0.413 | 0.413 |
| ETTm2 | 96 | **0.159** | **0.242** | 0.181 | 0.248 | 0.224 | 0.283 | 0.185 | 0.264 | 0.165 | 0.248 |
| | 192 | **0.216** | **0.283** | 0.253 | 0.296 | 0.308 | 0.335 | 0.257 | 0.311 | 0.225 | 0.295 |
| | 336 | **0.272** | **0.320** | 0.318 | 0.337 | 0.369 | 0.374 | 0.313 | 0.351 | 0.275 | 0.328 |
| | 720 | **0.354** | **0.375** | 0.417 | 0.396 | 0.460 | 0.430 | 0.402 | 0.408 | 0.367 | 0.385 |
| Weather | 96 | **0.144** | **0.181** | 0.183 | 0.216 | 0.206 | 0.220 | 0.172 | 0.218 | 0.149 | 0.198 |
| | 192 | **0.188** | **0.224** | 0.227 | 0.258 | 0.278 | 0.269 | 0.235 | 0.261 | 0.190 | 0.234 |
| | 336 | **0.238** | **0.267** | 0.286 | 0.297 | 0.335 | 0.312 | 0.296 | 0.305 | 0.248 | 0.279 |
| | 720 | **0.314** | **0.318** | 0.368 | 0.348 | 0.413 | 0.368 | 0.380 | 0.356 | 0.318 | 0.329 |

Table 5:    Full forecasting with input lengths $\{96, 336, 512\}$ and prediction lengths $\{96, 192, 336, 720\}$. The results are the best prediction across three input lengths.

## E    ABLATION ABOUT CROSS-MODEL FUSION

We conduct ablation experiments on each specific module within the cross-model fusion block, including current attention, memory attention, and gating fusion. As shown in Figure 7, compared to using only the PLM branch and CC-Time, we find that each module contributes uniquely to the overall performance. Among them, removing current attention has the most significant effect, indicating that the fusion of features from two types of models at the corresponding layers is crucial. Meanwhile, the memory attention ablation experiment shows that, in addition to fusion at the corresponding layers, by incorporating features from previous layers of the PLM branch, the effective fusion of the two models can be further enhanced, achieving better prediction accuracy.

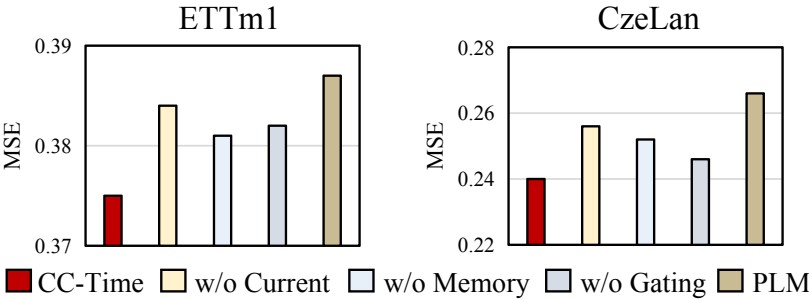

Figure 7: Each module ablation of cross-model fusion on the ETTm1 and CzeLan datasets. w/o current, w/o memory, and w/o gating represent removing current attention, memory attention, and gating fusion, respectively. PLM represents only the use of the PLM branch.

# F  MODEL ANALYSIS

## F.1  EFFECTIVENESS OF PRE-TRAINED LANGUAGE MODELS

To evaluate the effectiveness of pre-trained language models (PLMs), we conduct two types of experiments. In the first experiment, we consider three aspects: the number of PLM layers, parameter initialization, and whether to freeze PLM parameters. Specifically, CC-Time(3) and CC-Time(12) denote using 3 and 12 GPT layers, respectively, while the default CC-Time uses 6 layers. "Random init" refers to replacing pre-trained language model parameters with random initialization. "No freeze" represents that all PLM parameters are fine-tuned without freezing. As shown in Table 6, using too few or too many PLM layers decreases prediction accuracy. Too few layers may lead to insufficient extraction of complex features from PLMs, while too many layers can result in overly abstract features. Additionally, random parameter initialization significantly decreases accuracy, highlighting the importance of PLM knowledge in CC-Time for time series forecasting. Furthermore, not freezing PLM can cause catastrophic forgetting and overfitting, leading to poor prediction accuracy.

| Models | CC-Time | | CC-Time(3) | | CC-Time(12) | | Random init | | No freeze | |
|---|---|---|---|---|---|---|---|---|---|---|
| Metrics | MSE | MAE | MSE | MAE | MSE | MAE | MSE | MAE | MSE | MAE |
| ETTh1 | **0.424** | **0.424** | 0.438 | 0.429 | 0.437 | 0.431 | 0.445 | 0.438 | 0.452 | 0.447 |
| ETTh2 | **0.364** | **0.389** | 0.372 | 0.393 | 0.369 | 0.392 | 0.376 | 0.403 | 0.382 | 0.408 |

Table 6: Large language model analysis experiment.

In the second experiment, refer to Tan et al. (2024), we further validate the effectiveness of PLMs in CC-Time. Specifically, we focus on three aspects: removing the entire PLM layers of CC-Time (W/O PLM), replacing the PLM layers with a single layer of untrained Attention (LLM2Attn), replacing the PLM layers with a single layer of untrained Transformer (LLM2Trsf), and replacing the PLM layers with a single layer of untrained MLP (LLM2MLP). As shown in Table 7, CC-Time effectively utilizes LLM knowledge to model time series. When comparing W/O PLM with CC-Time, the results indicate that relying solely on basic time series modules like Patching and ReVIN is insufficient, and that effectively exploring the potential of PLM knowledge for time series forecasting is crucial. When comparing LLM2Attn and LLM2Trf with CC-Time, the results indicate that relying on single-layer attention or transformer for feature extraction is inadequate, and that fully leveraging the strengths of PLMs and time-series-specific models to capture time series features comprehensively is essential.

| Models | CC-Time | | W/O LLM | | LLM2Attn | | LLM2Trsf | | LLM2MLP | |
|---|---|---|---|---|---|---|---|---|---|---|
| Metrics | MSE | MAE | MSE | MAE | MSE | MAE | MSE | MAE | MSE | MAE |
| ETTh1 | **0.424** | **0.424** | 0.458 | 0.450 | 0.446 | 0.440 | 0.452 | 0.442 | 0.454 | 0.446 |
| ETTh2 | **0.363** | **0.389** | 0.385 | 0.406 | 0.388 | 0.409 | 0.378 | 0.401 | 0.383 | 0.402 |

Table 7: Performance of CC-Time with removing or replacing PLM layers.

### F.2 PARAMETER SENSITIVITY ANALYSIS

We conduct hyper-parameter sensitivity analysis of two key parameters in CC-Time on the ETTm1 dataset with five different random seeds: the loss weight $\lambda$, and the correlation weight $\epsilon$. Both the input length and prediction length are set to 96. As illustrated in Figure 8 (a), CC-Time achieves better prediction accuracy when $\lambda$ is set to 0.6. This suggests that the loss weight for the time series branch should be set relatively higher than that for the PLM Branch, as updating the time series branch also influences the corresponding PLM layers through cross-model fusion. Furthermore, Figure 8 (b) shows that CC-Time performs best when $\epsilon$ is set to 0.4, indicating that while balancing current and global correlation, it is beneficial to assign slightly more weight to focus on the current correlation. Overall, CC-Time exhibits relatively stable prediction accuracy with different values of the two hyperparameters.

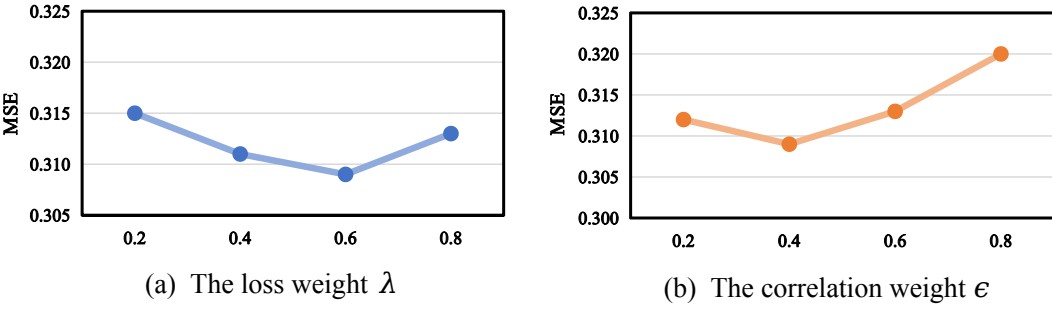

(a) The loss weight $\lambda$    (b) The correlation weight $\epsilon$

Figure 8: The effect of parameter $\lambda$ and parameter $\epsilon$. The results represent the mean across five independent runs with different random seeds, with observed variations bounded within $\pm 0.001$ for $\lambda$ and $\pm 0.0015$ for $\epsilon$.

### F.3 TEXT QUALITY ANALYSIS

To verify whether the PLM Branch in CC-Time truly understands the semantics of the channel text descriptions and is robust to text quality. We design three types of interventions: removing the channel text descriptions, replacing them with randomly generated text, and injecting noise into the constructed channel text descriptions. Note that all three interventions are applied to the entire channel text, including both the semantic descriptions and the statistical information. As shown in the Figure 9, injecting noise into the text leads to only a slight drop in prediction accuracy, indicating that CC-Time is robust to minor degradation in text quality. In contrast, replacing the channel descriptions with random text results in a significant performance drop—even worse than using no text at all—demonstrating that PLM Branch of the CC-Time understands and utilizes the semantic information in the channel descriptions to enhance the modeling of channel semantic correlations.

### F.4 TIME SERIES BRANCH REPLACEMENT ANALYSIS

To evaluate whether the PLM Branch and Cross-Model Fusion in CC-Time enhance other time series models, we replace its original time-series branch with two state-of-the-art time series models: HD-Mixer and PatchMixer. As shown in Table 8, the experimental results show consistent improvements

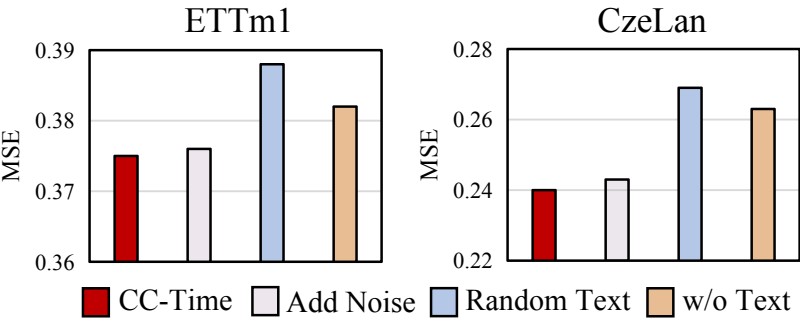

Figure 9: Channel text descriptions analysis on the ETTm1 and CzeLan datasets. w/o text represents removing channel text descriptions, Add Noise represents injecting noise into the constructed text descriptions, and Random Text represents randomly generating text descriptions. All three interventions are applied to the entire channel text, including both the semantic descriptions and the statistical information.

across three different datasets. This demonstrates that the proposed PLM Branch and Cross-Model Fusion effectively leverage the knowledge encoded in pre-trained language models and integrate it with time-series branches to comprehensively model temporal patterns, thereby improving forecasting performance. Moreover, the results highlight the strong generalizability of our method.

| Models | | HDMixer | | +CC-Time | | PatchMixer | | +CC-Time | |
|--------|-----|---------|-----|----------|-----|------------|-----|----------|-----|
| Metrics | | MSE | MAE | MSE | MAE | MSE | MAE | MSE | MAE |
| ETTm1 | 96 | 0.345 | 0.373 | 0.320 | 0.340 | 0.322 | 0.353 | 0.312 | 0.335 |
| | 192 | 0.380 | 0.391 | 0.364 | 0.370 | 0.363 | 0.380 | 0.358 | 0.367 |
| | 336 | 0.400 | 0.409 | 0.390 | 0.388 | 0.398 | 0.404 | 0.391 | 0.389 |
| | 720 | 0.466 | 0.443 | 0.450 | 0.432 | 0.455 | 0.439 | 0.448 | 0.427 |
| | avg | 0.397 | 0.404 | 0.381 | 0.382 | 0.384 | 0.394 | 0.377 | 0.379 |
| CzeLan | 96 | 0.203 | 0.256 | 0.190 | 0.237 | 0.204 | 0.251 | 0.189 | 0.232 |
| | 192 | 0.232 | 0.270 | 0.222 | 0.251 | 0.235 | 0.272 | 0.218 | 0.248 |
| | 336 | 0.255 | 0.296 | 0.249 | 0.273 | 0.267 | 0.294 | 0.253 | 0.276 |
| | 720 | 0.307 | 0.330 | 0.289 | 0.311 | 0.309 | 0.329 | 0.289 | 0.310 |
| | avg | 0.249 | 0.288 | 0.237 | 0.268 | 0.253 | 0.286 | 0.238 | 0.267 |
| Weather | 96 | 0.171 | 0.221 | 0.155 | 0.189 | 0.172 | 0.215 | 0.157 | 0.192 |
| | 192 | 0.223 | 0.263 | 0.208 | 0.242 | 0.219 | 0.252 | 0.202 | 0.235 |
| | 336 | 0.276 | 0.302 | 0.264 | 0.286 | 0.271 | 0.295 | 0.258 | 0.281 |
| | 720 | 0.345 | 0.347 | 0.340 | 0.334 | 0.349 | 0.344 | 0.342 | 0.336 |
| | avg | 0.253 | 0.283 | 0.241 | 0.263 | 0.252 | 0.276 | 0.239 | 0.261 |

Table 8: Time Series Branch Analysis. "+CC-Time" represents replacing the time series branch of CC-Time with other time series models.

### F.5 ABLATION ABOUT SEMANTIC DESCRIPTIONS AND STATISTICAL INFORMATION

To evaluate the individual contributions of the semantic information and statistical information in the channel text, we conduct a set of ablation studies, including: (1) removing both the semantic descriptions and statistical information (w/o Text), (2) removing only the semantic descriptions while retaining the statistical information (w/o semantic text), (3) removing only the statistical information while retaining the semantic descriptions (w/o statistical text). As shown in Table 9, both the seman-

| Models | CC-Time | | w/o Text | | w/o Semantic Text | | w/o statistical Text | |
|---|---|---|---|---|---|---|---|---|
| Metrics | MSE | MAE | MSE | MAE | MSE | MAE | MSE | MAE |
| ETTm1 | **0.375** | **0.377** | 0.382 | 0.382 | 0.380 | 0.379 | 0.378 | 0.379 |
| CzeLan | **0.240** | **0.261** | 0.263 | 0.277 | 0.252 | 0.270 | 0.248 | 0.266 |

Table 9: Performance of CC-Time with different text.

tic and statistical components contribute clear improvements to time series forecasting, enabling the model to capture channel correlations more comprehensively.

To more precisely assess the individual contributions of semantic and statistical information, we further inject light noise or random text separately into each part. As shown in the Table 10, CC-Time exhibits robustness to perturbations in both parts. When either the semantic description or the statistical information is replaced with random text, the prediction performance drops significantly, indicating that both parts of the information are important for prediction.

| Models | CC-Time | | Add noise of Semantic | | Random Semantic | | Add noise of Statistic | | Random statistic | |
|---|---|---|---|---|---|---|---|---|---|---|
| Metrics | MSE | MAE | MSE | MAE | MSE | MAE | MSE | MAE | MSE | MAE |
| ETTm1 | **0.375** | **0.377** | 0.377 | 0.378 | 0.387 | 0.395 | 0.378 | 0.379 | 0.384 | 0.391 |
| CzeLan | **0.240** | **0.261** | 0.243 | 0.265 | 0.258 | 0.273 | 0.244 | 0.263 | 0.252 | 0.272 |

Table 10: Analysis of the model's robustness for the semantic descriptions and statistical information.

## F.6 EFFICIENCY ANALYSIS

We conduct an efficiency comparison experiment between CC-Time and the baselines on the ETTh1 dataset, with both input and output lengths set to 96. We analyze the results from two aspects: training time per epoch and the number of trainable parameters. As illustrated in Figure 10, compared to PLM-based methods, such as Time-LLM and UniTime, CC-Time not only achieves superior prediction accuracy but also demonstrates significant advantages in training time and trainable parameters. In comparison to time-series-specific models, while iTransformer and PatchTST show better efficiency than CC-Time, their prediction accuracy is notably lower. Furthermore, in the few-shot forecasting scenario, where the training data is limited, the increase in CC-Time's cost compared to iTransformer and PatchTST is minor. Compared to Crossformer and FEDformer, CC-Time also exhibits advantages in efficiency. Overall, CC-Time effectively balances efficiency and prediction performance, and we believe that it is worthwhile to use PLMs to enhance prediction in CC-Time.

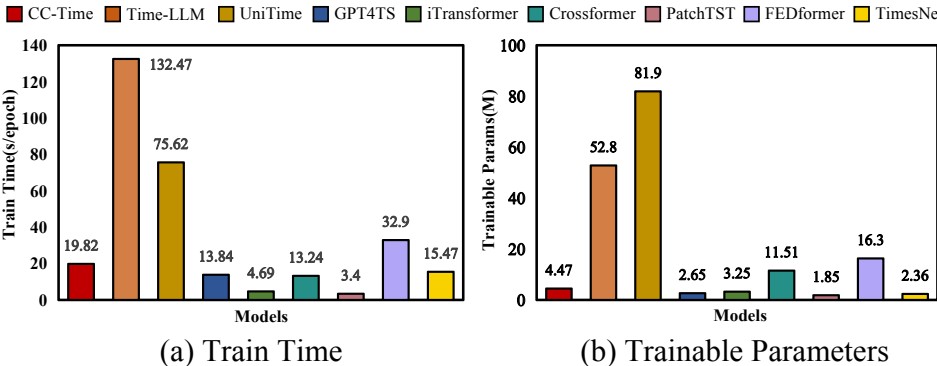

Figure 10: The efficiency analysis of CC-Time.

### F.6.1 INFERENCE EFFICIENCY

To better evaluate the inference efficiency of CC-Time, we compare it with several PLM-based models (Time-LLM, UniTime, and GPT4TS) as well as time series foundation models (MOIRAI, Chronos, and Timer). We adopt two evaluation metrics: maximum GPU memory usage and average inference time per batch. As shown in Table 11, CC-Time consistently demonstrates fast inference speed and low GPU memory consumption compared to both LLM-based models and time series foundation models.

|  | CC-Time | Time-LLM | UniTime | GPT4TS | MOIRAI Large | Chronos Large | Timer |
|---|---|---|---|---|---|---|---|
| GPU(MB) | 1261 | 8192 | 1878 | 1095 | 2009 | 10269 | 1435 |
| Inference(s) | 0.019 | 0.201 | 0.039 | 0.006 | 0.100 | 34.330 | 0.080 |

Table 11: The inference analysis of CC-Time and other baselines on the ETTh1 dataset with batch size of 1. We report the Max GPU Memory and Average Inference Time per batch.

### F.7 VARYING THE INPUT LENGTH

For time series forecasting, the input length determines the amount of historical information the model receives. To better assess the prediction accuracy of CC-Time, we select the top-performing models listed in Table 1 and conduct experiments using varying input lengths, specifically $T = \{48, 96, 192, 336, 512\}$. As illustrated in Figure 11, CC-Time consistently achieves state-of-the-art prediction accuracy across all input lengths. Notably, as the input length increases, the prediction metrics—MSE and MAE—demonstrate a clear downward trend. This reduction highlights CC-Time's ability to effectively model and leverage longer sequences, further showcasing its strength in capturing complex temporal dependencies.

## G  DISCUSSION

We provide a more systematic discussion of CC-Time and existing multimodal time series forecasting methods from the following aspects: 1) **Motivation and PLM role**: Unlike existing methods that focus on multimodal input fusion, CC-Time aims to unlock the potential of PLMs for time series forecasting, using PLMs to explicitly model temporal dependencies and channel correlations rather than merely as feature extractors (e.g., GPT4TS, S$^2$IP-LLM). 2) **Knowledge fusion**: Existing approaches typically fuse modalities only at the input or the intermediate layer (e.g., Time-LLM, TimeCMA). CC-Time uniquely performs adaptive fusion at each layer of the model. This layer-wise fusion allows PLMs and time-series models to jointly capture temporal patterns in a fine-grained manner. 3) **Input text design**: CC-Time constructs channel-level text descriptions, including semantic and statistical information, enhancing PLM modeling of complex temporal patterns. In contrast, existing methods mainly use global statistics for prompts, lacking channel-level semantic context (e.g., Time-LLM, S$^2$IP-LLM, TimeCMA). 4) **Prompt Length**: Like TimeCMA, CC-Time compresses the full text input into a single token that encapsulates its global semantic content, achieving both computational efficiency and comprehensive information retention.

## H  VISUALIZATION ANALYSIS

To assess the interpretability of the learned channel correlations, we conduct a visualization study comparing the model's learned correlation maps with ground-truth correlations. For a randomly selected sample from the Weather dataset, we compute its channel-wise Pearson correlation matrix as the local correlation, and we compute another correlation matrix over the entire training set as the global correlation. We then visualize the corresponding pre-Softmax attention maps from the PLM branch and the correlation extractor. As shown in Figure 12, the correlation map from the PLMs

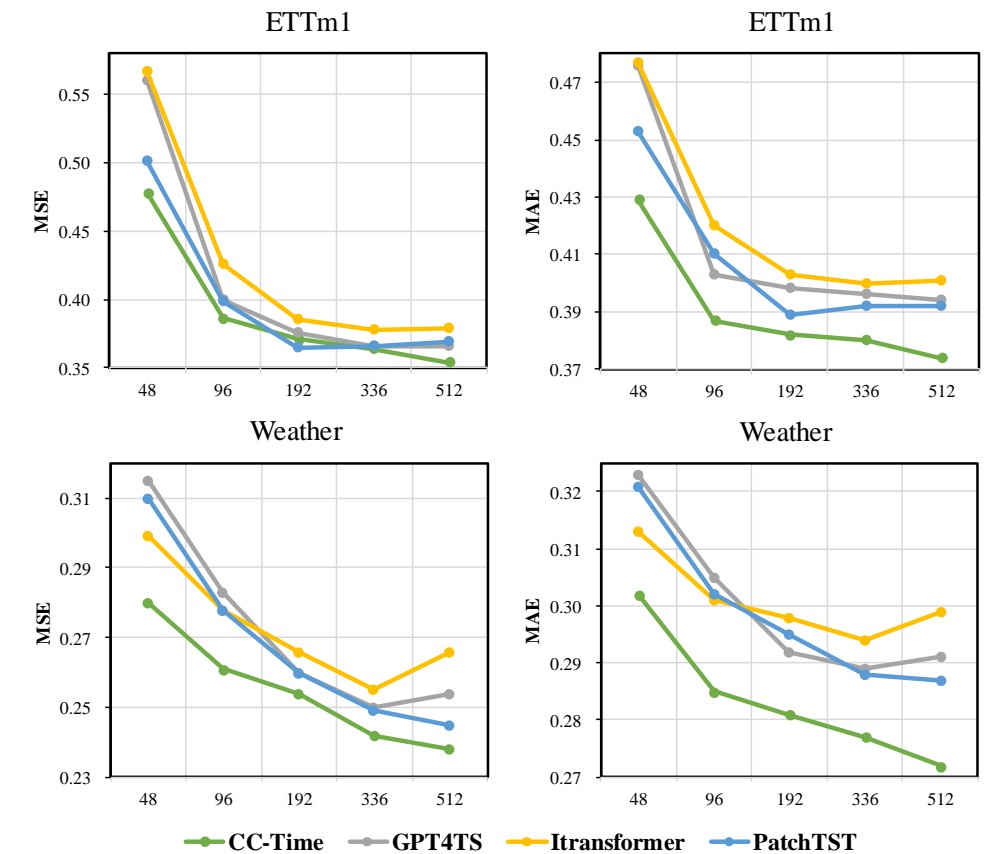

Figure 11: Results with different input lengths for the ETTm1 and Weather datasets. We set the input length $T = \{48, 96, 192, 336, 512\}$, the prediction length $F = 336$.

is similar to the sample-specific local correlation, while the extractor's map is similar to the global correlation of the full dataset. These results suggest that the PLMs capture more sample-specific channel dependencies, whereas the extractor captures more dataset-level correlations.

## I  FULL RESULTS

The full results about full forecasting and few-shot forecasting are provided in Tables 12 and 13.

## J  THE USE OF LARGE LANGUAGE MODELS (LLMs)

In CC-Time, we employ a pre-trained language model (GPT2) as the encoder in the PLM branch to extract features, thereby enhancing the model's capacity for time series understanding. It is worth emphasizing that LLMs were not involved in any part of the paper writing process

Table 12: Full forecasting results with the input length $T = 96$ and the prediction length $F = \{96, 192, 336, 720\}$.

| Models | | CC-Time | | FSCA | | S²IP-LLM | | Time-LLM | | UniTime | | GPT4TS | | PatchTST | | iTransformer | | Crossformer | | FEDformer | | TimesNet | |
|---|---|---|---|---|---|---|---|---|---|---|---|---|---|---|---|---|---|---|---|---|---|---|---|
| Metrics | | MSE | MAE | MSE | MAE | MSE | MAE | MSE | MAE | MSE | MAE | MSE | MAE | MSE | MAE | MSE | MAE | MSE | MAE | MSE | MAE | MSE | MAE |
| ETTm1 | 96 | 0.309 | 0.333 | 0.336 | 0.371 | 0.334 | 0.381 | 0.359 | 0.381 | 0.322 | 0.363 | 0.329 | 0.364 | 0.329 | 0.367 | 0.334 | 0.368 | 0.404 | 0.426 | 0.379 | 0.419 | 0.338 | 0.375 |
| | 192 | 0.354 | 0.363 | 0.376 | 0.391 | 0.377 | 0.382 | 0.383 | 0.393 | 0.366 | 0.387 | 0.368 | 0.382 | 0.367 | 0.385 | 0.377 | 0.391 | 0.450 | 0.451 | 0.426 | 0.441 | 0.374 | 0.387 |
| | 336 | 0.386 | 0.387 | 0.411 | 0.414 | 0.404 | 0.400 | 0.416 | 0.414 | 0.398 | 0.407 | 0.400 | 0.403 | 0.399 | 0.410 | 0.426 | 0.420 | 0.532 | 0.515 | 0.445 | 0.459 | 0.410 | 0.411 |
| | 720 | 0.451 | 0.426 | 0.470 | 0.449 | 0.469 | 0.437 | 0.483 | 0.449 | 0.454 | 0.440 | 0.460 | 0.439 | 0.454 | 0.439 | 0.491 | 0.459 | 0.666 | 0.589 | 0.543 | 0.490 | 0.478 | 0.450 |
| | Avg | 0.375 | 0.377 | 0.398 | 0.406 | 0.396 | 0.395 | 0.410 | 0.409 | 0.385 | 0.399 | 0.389 | 0.397 | 0.387 | 0.400 | 0.407 | 0.409 | 0.513 | 0.495 | 0.448 | 0.452 | 0.400 | 0.405 |
| ETTm2 | 96 | 0.171 | 0.248 | 0.181 | 0.261 | 0.177 | 0.261 | 0.193 | 0.280 | 0.183 | 0.266 | 0.178 | 0.263 | 0.175 | 0.259 | 0.180 | 0.264 | 0.287 | 0.366 | 0.203 | 0.287 | 0.187 | 0.267 |
| | 192 | 0.237 | 0.292 | 0.243 | 0.302 | 0.244 | 0.301 | 0.257 | 0.318 | 0.251 | 0.310 | 0.245 | 0.306 | 0.241 | 0.302 | 0.250 | 0.309 | 0.414 | 0.492 | 0.269 | 0.328 | 0.249 | 0.309 |
| | 336 | 0.296 | 0.330 | 0.303 | 0.339 | 0.306 | 0.343 | 0.317 | 0.353 | 0.319 | 0.351 | 0.309 | 0.347 | 0.305 | 0.343 | 0.311 | 0.348 | 0.597 | 0.542 | 0.325 | 0.366 | 0.321 | 0.351 |
| | 720 | 0.395 | 0.389 | 0.398 | 0.396 | 0.404 | 0.403 | 0.419 | 0.411 | 0.420 | 0.410 | 0.409 | 0.408 | 0.402 | 0.400 | 0.412 | 0.407 | 1.730 | 1.042 | 0.421 | 0.415 | 0.408 | 0.403 |
| | Avg | 0.274 | 0.314 | 0.281 | 0.324 | 0.282 | 0.327 | 0.296 | 0.340 | 0.293 | 0.334 | 0.285 | 0.331 | 0.280 | 0.326 | 0.288 | 0.332 | 0.757 | 0.610 | 0.304 | 0.349 | 0.291 | 0.332 |
| ETTh1 | 96 | 0.374 | 0.391 | 0.368 | 0.396 | 0.393 | 0.403 | 0.398 | 0.410 | 0.397 | 0.418 | 0.376 | 0.397 | 0.414 | 0.419 | 0.386 | 0.405 | 0.423 | 0.448 | 0.376 | 0.419 | 0.384 | 0.402 |
| | 192 | 0.414 | 0.417 | 0.425 | 0.426 | 0.443 | 0.429 | 0.442 | 0.435 | 0.434 | 0.439 | 0.438 | 0.426 | 0.460 | 0.445 | 0.441 | 0.436 | 0.471 | 0.474 | 0.420 | 0.448 | 0.436 | 0.429 |
| | 336 | 0.450 | 0.437 | 0.454 | 0.448 | 0.472 | 0.447 | 0.474 | 0.454 | 0.468 | 0.457 | 0.479 | 0.446 | 0.501 | 0.466 | 0.487 | 0.458 | 0.570 | 0.546 | 0.459 | 0.465 | 0.491 | 0.469 |
| | 720 | 0.449 | 0.454 | 0.476 | 0.480 | 0.480 | 0.478 | 0.471 | 0.472 | 0.469 | 0.477 | 0.495 | 0.476 | 0.500 | 0.488 | 0.503 | 0.491 | 0.653 | 0.621 | 0.506 | 0.507 | 0.521 | 0.500 |
| | Avg | 0.421 | 0.424 | 0.430 | 0.437 | 0.447 | 0.439 | 0.446 | 0.443 | 0.442 | 0.447 | 0.447 | 0.436 | 0.468 | 0.454 | 0.454 | 0.447 | 0.529 | 0.522 | 0.440 | 0.460 | 0.458 | 0.450 |
| ETTh2 | 96 | 0.282 | 0.330 | 0.294 | 0.346 | 0.309 | 0.364 | 0.295 | 0.346 | 0.296 | 0.345 | 0.295 | 0.348 | 0.302 | 0.348 | 0.297 | 0.349 | 0.745 | 0.584 | 0.358 | 0.397 | 0.340 | 0.374 |
| | 192 | 0.350 | 0.376 | 0.362 | 0.390 | 0.380 | 0.396 | 0.386 | 0.399 | 0.374 | 0.394 | 0.388 | 0.404 | 0.388 | 0.400 | 0.380 | 0.400 | 0.877 | 0.656 | 0.429 | 0.439 | 0.402 | 0.414 |
| | 336 | 0.405 | 0.419 | 0.383 | 0.408 | 0.422 | 0.432 | 0.447 | 0.443 | 0.415 | 0.427 | 0.421 | 0.435 | 0.426 | 0.433 | 0.428 | 0.432 | 1.043 | 0.731 | 0.496 | 0.487 | 0.452 | 0.452 |
| | 720 | 0.413 | 0.433 | 0.458 | 0.466 | 0.425 | 0.442 | 0.428 | 0.444 | 0.425 | 0.444 | 0.422 | 0.445 | 0.431 | 0.446 | 0.427 | 0.445 | 1.104 | 0.763 | 0.463 | 0.474 | 0.462 | 0.468 |
| | Avg | 0.362 | 0.389 | 0.374 | 0.402 | 0.384 | 0.408 | 0.389 | 0.408 | 0.377 | 0.402 | 0.381 | 0.408 | 0.386 | 0.406 | 0.383 | 0.406 | 0.942 | 0.683 | 0.437 | 0.449 | 0.414 | 0.427 |
| Weather | 96 | 0.157 | 0.192 | 0.176 | 0.216 | 0.181 | 0.220 | 0.195 | 0.233 | 0.171 | 0.214 | 0.177 | 0.218 | 0.174 | 0.214 | 0.158 | 0.230 | 0.217 | 0.296 | | | 0.172 | 0.220 |
| | 192 | 0.205 | 0.236 | 0.211 | 0.243 | 0.227 | 0.258 | 0.240 | 0.269 | 0.217 | 0.254 | 0.231 | 0.263 | 0.225 | 0.259 | 0.221 | 0.254 | 0.206 | 0.277 | 0.276 | 0.336 | 0.219 | 0.261 |
| | 336 | 0.261 | 0.280 | 0.278 | 0.296 | 0.280 | 0.298 | 0.293 | 0.306 | 0.274 | 0.293 | 0.283 | 0.300 | 0.278 | 0.297 | 0.278 | 0.296 | 0.272 | 0.335 | 0.339 | 0.380 | 0.280 | 0.306 |
| | 720 | 0.339 | 0.332 | 0.355 | 0.344 | 0.358 | 0.348 | 0.368 | 0.354 | 0.351 | 0.343 | 0.360 | 0.350 | 0.354 | 0.348 | 0.358 | 0.347 | 0.398 | 0.448 | 0.403 | 0.428 | 0.365 | 0.359 |
| | Avg | 0.240 | 0.260 | 0.255 | 0.274 | 0.261 | 0.281 | 0.274 | 0.290 | 0.253 | 0.276 | 0.264 | 0.284 | 0.258 | 0.280 | 0.257 | 0.277 | 0.258 | 0.315 | 0.309 | 0.360 | 0.259 | 0.287 |
| Electricity | 96 | 0.147 | 0.233 | 0.160 | 0.248 | 0.175 | 0.258 | 0.204 | 0.293 | 0.196 | 0.287 | 0.185 | 0.272 | 0.181 | 0.270 | 0.148 | 0.240 | 0.219 | 0.314 | 0.193 | 0.308 | 0.168 | 0.272 |
| | 192 | 0.165 | 0.247 | 0.172 | 0.260 | 0.182 | 0.270 | 0.207 | 0.295 | 0.199 | 0.291 | 0.199 | 0.276 | 0.188 | 0.274 | 0.162 | 0.253 | 0.231 | 0.324 | 0.201 | 0.315 | 0.184 | 0.289 |
| | 336 | 0.176 | 0.262 | 0.195 | 0.280 | 0.192 | 0.282 | 0.219 | 0.308 | 0.214 | 0.305 | 0.204 | 0.291 | 0.204 | 0.293 | 0.178 | 0.269 | 0.246 | 0.337 | 0.214 | 0.329 | 0.198 | 0.300 |
| | 720 | 0.209 | 0.289 | 0.221 | 0.306 | 0.245 | 0.322 | 0.263 | 0.341 | 0.254 | 0.335 | 0.245 | 0.324 | 0.246 | 0.324 | 0.225 | 0.317 | 0.280 | 0.363 | 0.246 | 0.355 | 0.220 | 0.320 |
| | Avg | 0.174 | 0.257 | 0.187 | 0.273 | 0.198 | 0.283 | 0.223 | 0.309 | 0.215 | 0.304 | 0.205 | 0.290 | 0.204 | 0.290 | 0.178 | 0.269 | 0.244 | 0.344 | 0.214 | 0.327 | 0.192 | 0.295 |
| Traffic | 96 | 0.400 | 0.243 | 0.455 | 0.287 | 0.466 | 0.300 | 0.536 | 0.359 | 0.458 | 0.301 | 0.462 | 0.307 | 0.462 | 0.295 | 0.395 | 0.268 | 0.522 | 0.290 | 0.587 | 0.366 | 0.593 | 0.321 |
| | 192 | 0.415 | 0.255 | 0.460 | 0.295 | 0.478 | 0.313 | 0.532 | 0.354 | 0.468 | 0.306 | 0.476 | 0.311 | 0.466 | 0.296 | 0.417 | 0.276 | 0.530 | 0.293 | 0.604 | 0.373 | 0.617 | 0.336 |
| | 336 | 0.432 | 0.264 | 0.472 | 0.306 | 0.489 | 0.320 | 0.530 | 0.349 | 0.485 | 0.310 | 0.488 | 0.317 | 0.482 | 0.304 | 0.433 | 0.283 | 0.558 | 0.305 | 0.621 | 0.383 | 0.629 | 0.336 |
| | 720 | 0.463 | 0.283 | 0.480 | 0.315 | 0.512 | 0.328 | 0.569 | 0.371 | 0.510 | 0.315 | 0.521 | 0.333 | 0.514 | 0.322 | 0.467 | 0.302 | 0.589 | 0.328 | 0.626 | 0.382 | 0.640 | 0.350 |
| | Avg | 0.427 | 0.262 | 0.466 | 0.300 | 0.486 | 0.315 | 0.541 | 0.358 | 0.480 | 0.308 | 0.488 | 0.317 | 0.481 | 0.304 | 0.428 | 0.282 | 0.549 | 0.304 | 0.610 | 0.376 | 0.619 | 0.335 |
| ZafNoo | 96 | 0.463 | 0.384 | 0.479 | 0.415 | 0.468 | 0.409 | 0.486 | 0.428 | 0.476 | 0.427 | 0.478 | 0.416 | 0.470 | 0.409 | 0.483 | 0.426 | 0.472 | 0.399 | 0.486 | 0.443 | 0.480 | 0.425 |
| | 192 | 0.524 | 0.422 | 0.558 | 0.458 | 0.557 | 0.461 | 0.561 | 0.467 | 0.554 | 0.468 | 0.561 | 0.462 | 0.545 | 0.450 | 0.548 | 0.457 | 0.520 | 0.432 | 0.569 | 0.485 | 0.554 | 0.465 |
| | 336 | 0.561 | 0.445 | 0.613 | 0.489 | 0.595 | 0.482 | 0.605 | 0.486 | 0.597 | 0.489 | 0.616 | 0.491 | 0.599 | 0.481 | 0.605 | 0.483 | 0.568 | 0.467 | 0.634 | 0.514 | 0.591 | 0.486 |
| | 720 | 0.634 | 0.481 | 0.707 | 0.532 | 0.691 | 0.531 | 0.713 | 0.541 | 0.698 | 0.527 | 0.721 | 0.542 | 0.692 | 0.525 | 0.670 | 0.518 | 0.642 | 0.498 | 0.795 | 0.607 | 0.722 | 0.540 |
| | Avg | 0.545 | 0.434 | 0.589 | 0.473 | 0.577 | 0.471 | 0.591 | 0.484 | 0.581 | 0.478 | 0.594 | 0.477 | 0.576 | 0.466 | 0.577 | 0.471 | 0.550 | 0.449 | 0.621 | 0.512 | 0.586 | 0.479 |
| CzeLan | 96 | 0.188 | 0.223 | 0.212 | 0.252 | 0.206 | 0.250 | 0.215 | 0.259 | 0.223 | 0.268 | 0.216 | 0.257 | 0.214 | 0.260 | 0.219 | 0.264 | 0.612 | 0.472 | 0.272 | 0.343 | 0.228 | 0.286 |
| | 192 | 0.220 | 0.245 | 0.251 | 0.282 | 0.238 | 0.275 | 0.255 | 0.293 | 0.245 | 0.290 | 0.247 | 0.276 | 0.243 | 0.279 | 0.250 | 0.285 | 0.776 | 0.540 | 0.307 | 0.363 | 0.255 | 0.301 |
| | 336 | 0.252 | 0.271 | 0.285 | 0.308 | 0.271 | 0.297 | 0.279 | 0.310 | 0.282 | 0.312 | 0.283 | 0.301 | 0.276 | 0.305 | 0.281 | 0.308 | 0.926 | 0.599 | 0.340 | 0.387 | 0.277 | 0.319 |
| | 720 | 0.301 | 0.307 | 0.347 | 0.356 | 0.335 | 0.339 | 0.339 | 0.350 | 0.348 | 0.352 | 0.348 | 0.346 | 0.341 | 0.349 | 0.347 | 0.354 | 1.344 | 0.730 | 0.402 | 0.424 | 0.315 | 0.346 |
| | Avg | 0.240 | 0.261 | 0.273 | 0.299 | 0.262 | 0.290 | 0.272 | 0.303 | 0.275 | 0.306 | 0.273 | 0.295 | 0.268 | 0.298 | 0.274 | 0.302 | 0.914 | 0.585 | 0.330 | 0.379 | 0.268 | 0.313 |

Table 13: 10% few shot forecasting results with the input length T = 96 and the prediction length F = $\{96, 192, 336, 720\}$.

| Models | | CC-Time | | S²IP-LLM | | FSCA | | Time-LLM | | UniTime | | GPT4TS | | PatchTST | | iTransformer | | Crossformer | | FEDformer | | TimesNet | |
|---|---|---|---|---|---|---|---|---|---|---|---|---|---|---|---|---|---|---|---|---|---|---|---|
| Metrics | | MSE | MAE | MSE | MAE | MSE | MAE | MSE | MAE | MSE | MAE | MSE | MAE | MSE | MAE | MSE | MAE | MSE | MAE | MSE | MAE | MSE | MAE |
| ETTm1 | 96 | 0.338 | 0.358 | 0.358 | 0.381 | 0.371 | 0.391 | 0.366 | 0.379 | 0.366 | 0.386 | 0.350 | 0.373 | 0.354 | 0.378 | 0.379 | 0.392 | 0.476 | 0.486 | 0.435 | 0.483 | 0.475 | 0.444 |
| | 192 | 0.382 | 0.383 | 0.402 | 0.404 | 0.400 | 0.400 | 0.404 | 0.400 | 0.405 | 0.406 | 0.396 | 0.393 | 0.396 | 0.399 | 0.423 | 0.414 | 0.581 | 0.563 | 0.479 | 0.511 | 0.586 | 0.484 |
| | 336 | 0.415 | 0.408 | 0.440 | 0.428 | 0.438 | 0.429 | 0.434 | 0.416 | 0.437 | 0.427 | 0.429 | 0.418 | 0.429 | 0.421 | 0.464 | 0.440 | 0.708 | 0.646 | 0.562 | 0.551 | 0.517 | 0.467 |
| | 720 | 0.490 | 0.449 | 0.489 | 0.454 | 0.509 | 0.463 | 0.492 | 0.457 | 0.489 | 0.456 | 0.484 | 0.449 | 0.506 | 0.469 | 0.537 | 0.486 | 0.777 | 0.666 | 0.701 | 0.630 | 0.582 | 0.512 |
| | Avg | 0.406 | 0.399 | 0.422 | 0.416 | 0.429 | 0.421 | 0.424 | 0.413 | 0.424 | 0.419 | 0.415 | 0.408 | 0.421 | 0.416 | 0.450 | 0.431 | 0.635 | 0.590 | 0.544 | 0.544 | 0.540 | 0.476 |
| ETTm2 | 96 | 0.183 | 0.266 | 0.192 | 0.272 | 0.194 | 0.273 | 0.198 | 0.286 | 0.202 | 0.287 | 0.191 | 0.279 | 0.193 | 0.281 | 0.194 | 0.280 | 0.672 | 0.582 | 0.234 | 0.322 | 0.202 | 0.285 |
| | 192 | 0.255 | 0.315 | 0.259 | 0.318 | 0.262 | 0.320 | 0.262 | 0.322 | 0.265 | 0.324 | 0.260 | 0.323 | 0.260 | 0.325 | 0.261 | 0.321 | 1.225 | 0.819 | 0.310 | 0.374 | 0.299 | 0.354 |
| | 336 | 0.315 | 0.351 | 0.325 | 0.36 | 0.328 | 0.360 | 0.328 | 0.361 | 0.322 | 0.358 | 0.325 | 0.363 | 0.325 | 0.364 | 0.325 | 0.364 | 1.135 | 0.788 | 0.383 | 0.424 | 0.336 | 0.371 |
| | 720 | 0.414 | 0.411 | 0.457 | 0.435 | 0.455 | 0.434 | 0.436 | 0.420 | 0.421 | 0.414 | 0.425 | 0.418 | 0.423 | 0.421 | 0.440 | 0.428 | 1.873 | 1.032 | 0.558 | 0.529 | 0.482 | 0.455 |
| | Avg | 0.291 | 0.335 | 0.308 | 0.346 | 0.310 | 0.347 | 0.306 | 0.347 | 0.303 | 0.346 | 0.300 | 0.345 | 0.300 | 0.347 | 0.305 | 0.349 | 1.226 | 0.805 | 0.371 | 0.412 | 0.329 | 0.366 |
| ETTh1 | 96 | 0.390 | 0.403 | 0.415 | 0.418 | 0.420 | 0.424 | 0.404 | 0.424 | 0.423 | 0.424 | 0.399 | 0.415 | 0.398 | 0.407 | 0.514 | 0.481 | 0.555 | 0.554 | 0.519 | 0.522 | 0.606 | 0.533 |
| | 192 | 0.448 | 0.443 | 0.459 | 0.439 | 0.469 | 0.450 | 0.458 | 0.446 | 0.467 | 0.448 | 0.454 | 0.441 | 0.463 | 0.447 | 0.603 | 0.524 | 0.601 | 0.581 | 0.600 | 0.562 | 0.743 | 0.588 |
| | 336 | 0.479 | 0.452 | 0.504 | 0.469 | 0.510 | 0.465 | 0.496 | 0.465 | 0.503 | 0.461 | 0.502 | 0.469 | 0.515 | 0.470 | 0.683 | 0.562 | 0.880 | 0.707 | 0.670 | 0.574 | 0.918 | 0.643 |
| | 720 | 0.522 | 0.495 | 0.53 | 0.505 | 0.536 | 0.508 | 0.560 | 0.520 | 0.535 | 0.504 | 0.526 | 0.504 | 0.541 | 0.508 | 0.842 | 0.638 | 1.303 | 0.916 | 0.609 | 0.544 | 0.873 | 0.638 |
| | Avg | 0.459 | 0.448 | 0.477 | 0.457 | 0.484 | 0.461 | 0.479 | 0.462 | 0.482 | 0.459 | 0.470 | 0.457 | 0.479 | 0.458 | 0.660 | 0.551 | 0.834 | 0.689 | 0.600 | 0.551 | 0.785 | 0.600 |
| ETTh2 | 96 | 0.290 | 0.337 | 0.305 | 0.35 | 0.355 | 0.389 | 0.302 | 0.348 | 0.323 | 0.363 | 0.304 | 0.351 | 0.304 | 0.348 | 0.334 | 0.375 | 1.236 | 0.809 | 0.384 | 0.428 | 0.389 | 0.419 |
| | 192 | 0.389 | 0.397 | 0.408 | 0.412 | 0.425 | 0.427 | 0.391 | 0.400 | 0.407 | 0.413 | 0.404 | 0.411 | 0.414 | 0.410 | 0.429 | 0.430 | 1.102 | 0.798 | 0.462 | 0.470 | 0.496 | 0.471 |
| | 336 | 0.478 | 0.455 | 0.478 | 0.462 | 0.480 | 0.465 | 0.470 | 0.462 | 0.481 | 0.470 | 0.465 | 0.457 | 0.470 | 0.456 | 0.479 | 0.466 | 1.243 | 0.863 | 0.466 | 0.481 | 0.526 | 0.495 |
| | 720 | 0.492 | 0.477 | 0.476 | 0.47 | 0.510 | 0.489 | 0.478 | 0.472 | 0.490 | 0.481 | 0.504 | 0.481 | 0.505 | 0.485 | 0.500 | 0.485 | 1.320 | 0.910 | 0.456 | 0.481 | 0.510 | 0.491 |
| | Avg | 0.412 | 0.416 | 0.416 | 0.423 | 0.443 | 0.443 | 0.410 | 0.420 | 0.425 | 0.431 | 0.419 | 0.425 | 0.423 | 0.424 | 0.435 | 0.439 | 1.225 | 0.845 | 0.442 | 0.465 | 0.480 | 0.469 |
| Weather | 96 | 0.175 | 0.210 | 0.184 | 0.229 | 0.181 | 0.227 | 0.195 | 0.233 | 0.193 | 0.233 | 0.192 | 0.230 | 0.185 | 0.225 | 0.191 | 0.230 | 0.425 | 0.493 | 0.334 | 0.401 | 0.198 | 0.246 |
| | 192 | 0.224 | 0.254 | 0.232 | 0.265 | 0.230 | 0.264 | 0.241 | 0.270 | 0.237 | 0.268 | 0.237 | 0.267 | 0.238 | 0.263 | 0.238 | 0.269 | 0.635 | 0.620 | 0.389 | 0.430 | 0.243 | 0.279 |
| | 336 | 0.282 | 0.295 | 0.292 | 0.314 | 0.285 | 0.305 | 0.290 | 0.304 | 0.289 | 0.304 | 0.290 | 0.303 | 0.296 | 0.303 | 0.293 | 0.307 | 0.540 | 0.570 | 0.444 | 0.463 | 0.295 | 0.313 |
| | 720 | 0.356 | 0.343 | 0.365 | 0.351 | 0.364 | 0.353 | 0.364 | 0.353 | 0.362 | 0.351 | 0.363 | 0.352 | 0.368 | 0.351 | 0.369 | 0.356 | 0.775 | 0.688 | 0.559 | 0.531 | 0.363 | 0.360 |
| | Avg | 0.259 | 0.275 | 0.268 | 0.289 | 0.265 | 0.288 | 0.273 | 0.290 | 0.270 | 0.289 | 0.270 | 0.288 | 0.271 | 0.285 | 0.272 | 0.290 | 0.593 | 0.592 | 0.432 | 0.456 | 0.274 | 0.299 |

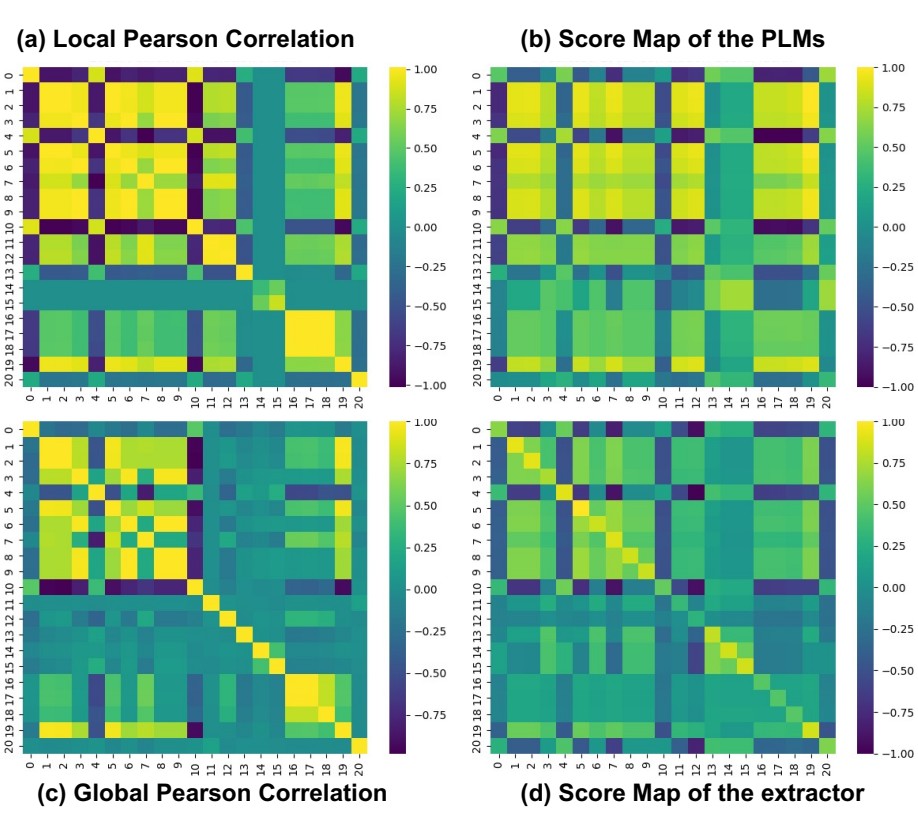

Figure 12: Visualization of channel correlations learned by the PLMs and the Correlation Extractor.

