# OpenReview forum: "CC-Time: Cross-Model and Cross-Modality Time Series Forecasting"
_ICLR.cc/2026/Conference — Submitted to ICLR 2026_

### Official Review · Reviewer_QNfh · 2025-10-15

**Soundness:** 3
**Presentation:** 3
**Contribution:** 3
**Rating:** 8
**Confidence:** 4

**Summary:**

This paper proposes CC-Time, a Cross-Model and Cross-Modality framework for time series forecasting that integrates the strengths of pre-trained language models and time-series-specific models. Traditional forecasting approaches either focus on numerical modeling of temporal patterns or exploit PLMs for sequence understanding, but each alone is limited. CC-Time bridges this gap by combining both paradigms and leveraging semantic knowledge from text to model complex temporal and channel correlations.The framework contains a PLM branch, a time-series branch, and a cross-model fusion module. The PLM branch introduces a cross-modality modeling mechanism, combining time-series data with automatically generated channel text descriptions to help PLMs capture both temporal dependencies and semantic relationships among channels. The time-series branch, built on a transformer structure, models fine-grained numerical temporal dynamics. The cross-model fusion block adaptively integrates multi-level features from both branches using attention and gating mechanisms, yielding a unified representation that captures both semantic and quantitative aspects of time series data.
Experiments conducted on real-world datasets from diverse domains such as energy, weather, and traffic show that CC-Time achieves state-of-the-art prediction accuracy under both full-data and few-shot settings.

**Strengths:**

This paper is original in proposing a cross-model and cross-modality framework that unites pre-trained language models with time-series-specific architectures for forecasting. The idea of using automatically generated channel text descriptions to inject semantic knowledge into time-series modeling is interesting. The paper is well-structured. Its comprehensive experiments across multiple datasets and settings demonstrates strong generalization and robustness.  In terms of clarity, the paper is clearly written, well-organized, and supported by informative figures that make the architecture and motivation easy to follow. In all, this paper opens new directions for multimodal and cross-domain modeling in time-series research.

**Weaknesses:**

* The paper does not provide a detailed hyperparameter sensitivity analysis, leaving some uncertainty about the robustness and stability of the proposed framework across different parameter settings.
* The paper demonstrates that CC-Time captures richer correlations, but remains a little bit unclear how these align with real-world semantics or physical dependencies among channels.

**Questions:**

* What is the computational overhead of the two branches？

* Since the model relies on automatically generated channel text descriptions, how does the quality of these descriptions affect performance?

---

> ### Author Response · Authors · 2025-11-21
> **Response to Reviewer QNfh**
>
> Many Thanks to Reviewer QNfh for providing a detailed review and insightful questions about sensitivity analysis, computational overhead analysis, and the channel text quality analysis. We have revised our paper accordingly.
>
> **W1:** Hyperparameter sensitivity analysis
>
> In the original paper, we conduct the hyperparameter sensitivity analysis for the loss weight λ and the correlation weight ϵ $\underline { \text {in Appendix F.2}}$.
>
> |  $\lambda = 0$ | $\lambda = 0$.2 | $\lambda = 0$.4 | $\lambda = 0$.6 | $\lambda = 0$.8 |
> | --- | --- | --- | --- | --- |
> | MSE | MSE  | MSE   | MSE  | MSE  |
> | 0.317 | 0.315   | 0.311  | 0.309  | 0.313  |
>
>
> | $\epsilon = 0$.2 | $\epsilon = 0$.4 | $\epsilon = 0$.6 | $\epsilon = 0$.8 |
> | --- | --- | --- | --- |
> | MSE  | MSE   | MSE  | MSE  |
> | 0.312   | 0.309  | 0.314  | 0.320  |
>
>
> **W2:** Channel semantic correlation modeling
>
> When modeling channel correlations, CC-Time takes both channel text information and the raw time series as input. The channel text includes semantic descriptions generated by LLMs as well as overall statistical information for each channel. Since PLMs are pre-trained on large-scale corpora and have acquired extensive knowledge, they can interpret the semantic information from the channel text while simultaneously modeling the numerical correlations among channels. The combination of these two sources allows PLMs to capture more complete channel correlations that better reflect real-world semantics and physical dependencies.
>
>
>
> **Q1:** Computational overhead analysis
>
> **Computational Cost Analysis:**
>
> * **PLM Branch:**
>
>   * *PLM Channel Layer:* receives input $E_{cross} \in \mathbb{R}^{C \times D_l}$, where the number of tokens equals the number of channels C. Its computational complexity is $O(C^2 \cdot D_l)$.
>   * *PLM Temporal Layer:* receives input $E_{patch} \in \mathbb{R}^{C \times N_P \times D_l}$, where the number of tokens equals the number of patches $N_P$. Its complexity is $O(N_P^2 \cdot D_l)$.
>
> * **Time Series Branch:** It receives input $E_{patch}^{\prime} \in \mathbb{R}^{C \times N_P \times D_t}$ with $N_P$ tokens. Its computational complexity is $O(N_P^2 \cdot D_t)$.
>
> Since the number of patches $N_P$ and channels $C$ are relatively small, the overall theoretical complexity of the model remains moderate.
>
>
>
> **Efficiency Analysis**: In the original paper, we connduct the efficiency analysis in Appendix F.5, including model parameter count, training time, GPU usage, and inference time. Compared with existing PLM-based methods and time-series–specific models, CC-Time demonstrates competitive training and inference efficiency.
>
> **Q2:** Channel text quality analysis
>
> $\underline{\text{Figure 6 of the Appendix}}$ shows that, by leveraging the extensive knowledge in LLMs along with our prompts (which include dataset domain information and specific channel names), the LLMs can generate accurate channel semantic information, providing a detailed explanation for each channel.
>
> To assess the impact of text quality, we conduct experiments $\underline{\text{ in Appendix F.3}}$. As shown in $\underline{\text{Figure 9}}$, adding noise to the text leads to only a slight drop in prediction accuracy, indicating that CC-Time is robust to minor degradation in channel text quality.

---

### Official Review · Reviewer_NQT5 · 2025-10-29

**Soundness:** 3
**Presentation:** 3
**Contribution:** 2
**Rating:** 4
**Confidence:** 4

**Summary:**

This paper introduces CC-Time, a dual-branch architecture that integrates large language models (LLMs) and time-series-specific models for time series forecasting (TSF). The model addresses two key questions: (1) how to enable pre-trained language models (PLMs) to jointly capture temporal dependencies and variable correlations, and (2) how to leverage the complementary strengths of LLMs and traditional time-series models through adaptive fusion.To this end, the authors propose Cross-Modality Learning to enhance correlation modeling between numerical and textual representations, and a Cross-Level Fusion (CLF) block to integrate features from different representational levels. Experiments on nine real-world datasets show consistent performance improvements in both full-data and few-shot settings.

**Strengths:**

1. Novel integration of PLMs and TS models: The paper presents a thoughtful framework that unites semantic (LLM-based) and numerical (Transformer-based) modeling. The design of the CLF block reflects careful consideration of cross-representational learning in time series forecasting.

2. Innovative cross-modality correlation modeling: Incorporating text-based variable descriptions and a correlation extractor allows the model to capture both global and local dependencies from semantic and numeric perspectives.

3. Strong empirical validation: Experiments cover a broad range of datasets and forecasting horizons, consistently showing superior results over LLM-based and TS-specific baselines. Few-shot evaluations further demonstrate the potential of LLMs for low-data forecasting.

4. Comprehensive component analysis: The paper provides ablation studies on cross-level fusion and cross-modality correlation modules, along with model-depth and freezing analyses that strengthen the empirical soundness of the claims.

**Weaknesses:**

1. Lack of explicit modality alignment between time-series embeddings and PLM semantic space: The paper directly feeds time-series embeddings into the pre-trained PLM without introducing any explicit alignment constraint between the numerical and linguistic modalities. This raises concerns about whether the frozen PLM can effectively interpret unaligned numeric encodings, especially since no contrastive or projection-based objective is applied to bridge the representational gap. As a result, the semantic structure within the PLM may not correspond to the statistical dynamics of the time-series features, potentially limiting the PLM’s contribution during fusion.

2. Limited interpretability of the correlation modeling process: While the paper proposes both a correlation extractor and PLM-based correlation layers to capture global and local dependencies across variables, the modeling results are not visually or quantitatively analyzed.

3. Insufficient ablation to isolate the PLM branch’s contribution: The proposed Cross-Model Fusion integrates semantic correlations from the PLM with numerical representations from the time-series model. However, the paper lacks an ablation experiment isolating the PLM branch’s effect, which is essential to determine whether the semantic representations learned by the PLM genuinely enhance forecasting performance. Without such analysis, it is unclear whether the observed gains primarily come from the PLM, the time-series backbone, or their interaction. This omission weakens the causal interpretability of the architecture’s design claims.

**Questions:**

1. Would it be beneficial to introduce an explicit alignment objective (e.g., contrastive loss or learned projection) to map time-series embeddings into the PLM’s semantic space? If the PLM remains largely frozen, how does it meaningfully process such unaligned numerical representations?

2. Can the authors provide visualization or quantitative analysis (e.g., inter-channel correlation maps or attention weight distributions) to validate the correctness and interpretability of the extracted correlations?

3. Could the authors include a control experiment where the PLM branch is removed or replaced with a lightweight MLP to confirm that the semantic correlations extracted by the PLM provide measurable improvements to the TS branch’s predictive accuracy?

---

> ### Author Response · Authors · 2025-11-21
> **Response to Reviewer NQT5**
>
> We would like to sincerely thank Reviewer NQT5 for providing the detailed review and insightful suggestions regarding the explicit modality alignment, visualization analysis, and ablation experiment about PLMs. We have revised our paper accordingly.
>
> **W1, Q1:** Explicit modality alignment between the time series embeddings and PLM semantic space
>
> We would like to clarify that CC-Time explicitly aligns the time series with the semantic space of the pretrained language model (PLM), through the following mechanisms:
>
> - **Learnable projections**: In the PLM Branch, both the Patch Embedding and Cross-Modality Embedding include learnable projection layers that map the time series data to the required dimensions of the PLM.
>
> - **Partial fine-tuning of the PLMs**: The PLMs is not completely frozen. Following GPT4TS [1], we freeze the main parameters while fine-tuning the positional encoding and layer normalization. This partial fine-tuning effectively aligns the time-series embeddings with the PLM semantic space.
> In the original paper, we conduct experiments with randomly initialized PLMs and fully frozen PLMs in the $\underline{\text{Appendix F.1}}$, both of which resulted in decreased forecasting performance compared with partial fine-tuning. This suggests that only fine-tuning the key components of the PLMs could well align time series and the PLM's semantic space.
>
> - **Cross-Model Fusion**: Components such as Current Attention and Memory Attention fuse representations from the PLM Branch and the Time Series Branch, indirectly assisting the alignment of the PLM Branch with the time series embeddings.
>
> [1] One Fits All: Power General Time Series Analysis by Pretrained LM
>
> **W2, Q2:** Visualization analysis about channel correlations
>
> To assess the interpretability of the learned channel correlations, we add the visualization of the correlation maps from the PLMs and the Correlation Extractor $\underline{\text{ in the Appendix H of the revised paper}}$. As shown in $\underline{\text{the Figure 12 of the revised paper }}$, the correlation map from the PLMs is similar to the sample local correlation, and the correlation map from the Correlation Extractor is similar to the global correlation of the full dataset. This suggests that the PLMs capture more sample-specific channel dependencies, whereas the extractor captures more dataset-level correlations.
>
> **W3, Q3:** Ablation experiment about removing PLMs or replacing PLMs
>
> $\underline{\text {In Table 7 of the original paper}}$, we have conducted an effectiveness analysis of the PLMs, including: removing all PLM layers from CC-Time (W/O PLM), replacing the PLM layers with a single untrained Attention layer (LLM2Attn), and replacing them with a single untrained Transformer layer (LLM2Trsf). As suggested, we also add an additional experiment replacing the PLM layers with an MLP (LLM2MLP).
>
> As shown in the table, removing the PLM layers or replacing them with an MLP or Attention leads to a performance drop, indicating that CC-Time effectively leverages the knowledge within the PLM and unlocks its potential for time series modeling.
>
> |  Models | CC-Time | W/O PLM | LLM2Attn | LLM2Trsf | LLM2MLP |
> | --- | --- | --- | --- | --- | --- |
> | Metrics | MSE / MAE | MSE / MAE | MSE / MAE | MSE / MAE  | MSE / MAE |
> | ETTh1 | 0.424 / 0.424 | 0.458 / 0.450 | 0.446 / 0.440  | 0.452 / 0.442 | 0.454 / 0.446 |
> | ETTh2 | 0.363 / 0.389  | 0.385 / 0.406 | 0.388 / 0.409 | 0.378 / 0.401 | 0.383 / 0.402 |

---

> > ### Comment · Reviewer_NQT5 · 2025-11-27
> >
> > The author addresses some of my main concerns, therefore, I increase my rating to wa accordingly.

---

> ### Author Response · Authors · 2025-11-28
>
> We would like to thank Reviewer NQT5 for providing a detailed valuable review, which helps us a lot in the rebuttal and paper revision.
>
> Thanks again for your response and for raising the score!

---

### Official Review · Reviewer_xBLS · 2025-10-30

**Soundness:** 3
**Presentation:** 3
**Contribution:** 3
**Rating:** 6
**Confidence:** 5

**Summary:**

This paper works on time series forecasting (TSF) and presents CC-Time, a dual-branch framework integrating LLM and traditional TSF models. CC-Time addresses two core questions: which time series features PLMs can model, and whether PLMs alone suffice for TSF.
CC-Time adopts cross-modality learning, which combines time series and text (that describe channels), to capture temporal dependencies and channel correlations. Further, it introduces cross-model fusion (CMF) block to adaptively integrate knowledge from both branches. Extensive experiments have been conducted to validate the effectiveness of proposed modules.

**Strengths:**

1. It's reasonable to incorporate the ability of LLMs into traditional TSF models in a multimodal manner.
2. It's interesting to use ChatGPT to describe each channel, making the essence and functionality of each channel more clear, which may help to better model channel-wise correlations and improve interpretability.
3. The CMF block seems to be novel and reasonable.
4. Extensive experiments have been conducted to validate the effectiveness of modules. Particularly, this paper works on full-data and few-shot settings to provide robust evaluation.

**Weaknesses:**

1. A discussion between CC-Time and existing multimodal TSF methods (that also use both time series and textual data) is strongly recommended, which would make the contribution of this work more prominent.
- What are the differences between the constructed textual input, in terms of both method and motivation.
- Prompt length.
- Why such textual input can make your multimodal data fusion unique (compared to existing methods like Time-LLM, TimeCMA).
2. The computational cost of each module (particularly the attention-related modules), and efficiency analysis are recommended.
3. In Table 3 and Appendix F.1, the reasons why introducing larger LLMs cannot further boost the performance of TSF is not clear.
4. This paper captures channel-wise correlations near the input end, and TimeCMA puts similar-purpose module after data fusion (i.e., near the output end), an experiment (conducted under the same conditions) studying the positions of channel-wise correlation capturing is missing.

**Questions:**

N.A.

---

> ### Author Response · Authors · 2025-11-21
> **Response to Reviewer xBLS**
>
> We would like to sincerely thank Reviewer xBLS for providing the insightful review about the discussion with existing methods, computation cost analysis, and large LLMs' performance. We have revised our paper accordingly.
>
> **W1:** Discussion with existing multimodal time series forecasting methods
>
> Thank you for the valuable comments. Based on your suggestions, we provide a more systematic discussion of CC-Time compared with existing multimodal time series forecasting methods:
>
> - **Motivation and PLM role**: Unlike existing methods that focus on multimodal input fusion, CC-Time aims to unlock the potential of PLMs for time series forecasting, using PLMs to explicitly model temporal dependencies and channel correlations rather than merely as feature extractors (e.g., GPT4TS, S$^2$IP-LLM).
>
> - **Textual input**: CC-Time constructs channel-level text descriptions, including semantic and statistical information, enhancing PLM modeling of complex temporal patterns. In contrast, existing methods mainly use global statistics for prompts, lacking channel-level semantic context (e.g., Time-LLM, S$^2$IP-LLM, TimeCMA).
>
> - **Prompt Length**: Like TimeCMA, CC-Time compresses the full text input into a single token that encapsulates its global semantic content, achieving both computational efficiency and comprehensive information retention.
>
>
> - **Knowledge fusion**: Existing approaches typically fuse modalities only at the input or the intermediate layer (e.g., Time-LLM, TimeCMA). CC-Time uniquely performs adaptive fusion at each layer of the model. This layer-wise fusion allows PLMs and time-series models to jointly capture temporal patterns in a fine-grained manner.
>
>
> We also add the above discussion in $\underline{\text{Section G of the revised supplementary}}$.
>
> **W2:** The computational cost and efficiency analysis
>
> **Computational Cost Analysis:**
>
> * **PLM Branch:**
>
>   * *PLM Channel Layer:* receives input $E_{cross} \in \mathbb{R}^{C \times D_l}$, where the number of tokens equals the number of channels C. Its computational complexity is $O(C^2 \cdot D_l)$.
>   * *PLM Temporal Layer:* receives input $E_{patch} \in \mathbb{R}^{C \times N_P \times D_l}$, where the number of tokens equals the number of patches $N_P$. Its complexity is $O(N_P^2 \cdot D_l)$.
>
> * **Time Series Branch:** receives input $E_{patch}^{\prime} \in \mathbb{R}^{C \times N_P \times D_t}$ with $N_P$ tokens. Its computational complexity is $O(N_P^2 \cdot D_t)$.
>
> * **Cross-Model Fusion:** Both Current Attention and Memory Attention operate on features of dimension $\mathbb{R}^{C \times N_m \times D_l}$, where the number of tokens $N_m$ is similar to $N_P$. The complexity is $O(N_m^2 \cdot D_l)$.
>
> Since the number of patches $N_P$ and channels $C$ are relatively small, the overall theoretical complexity of the model remains moderate.
>
>
>
> **Efficiency Analysis**: In the original paper, we have conducted the efficiency analysis $\underline{\text{in Appendix F.6}}$, including model parameter count, training time, GPU usage, and inference time. Compared with existing PLM-based methods and time-series–specific models, CC-Time demonstrates competitive training and inference efficiency.
>
>
> **W3:** Large LLMs' performance on time series forecasting
>
> In Table 3, for the LLaMA models ranging from 7B to 13B, the forecasting performance remains stable. This suggests that CC-Time may not heavily rely on the additional capabilities of larger models (e.g., reasoning), but instead primarily benefits from their semantic understanding and representation learning.
>
> In Appendix F.1, we observe that using a 6-layer GPT-2 outperforms the 12-layer. The phenomenon occurs in models such as GPT4TS and S$^2$IP-LLM, which may be due to deeper layers introducing task-irrelevant semantics or knowledge that interfere with the modeling of temporal patterns.
>
> **W4:** The difference in modeling channel correlation between CC-Time and TimeCMA
>
> The difference between CC-Time and TimeCMA in channel modeling is not in the position of channel-wise correlation, but in how it is modeled.
> CC-Time primarily leverages the modeling capability of the pre-trained language model along with channel text descriptions to capture complex channel correlations, such as semantic correlations. TimeCMA relies on a multivariate Transformer Decoder to capture channel correlations mainly from numerical time series data.
>
> To further validate this distinction, we remove the PLM Channel Layer in CC-Time and add channel modeling attention in the Time Series Branch (Variant). The experiment results are shown below:
>
>
> |  | CC-Time  | Variant  |
> | --- | --- | --- |
> | Metrics | MSE / MAE | MSE / MAE |
> | ETTm1 | 0.375 / 0.377 | 0.386 / 0.396 |
> | CzeLan | 0.240 / 0.261 | 0.264 / 0.285 |

---

### Official Review · Reviewer_L1sA · 2025-10-31

**Soundness:** 2
**Presentation:** 1
**Contribution:** 2
**Rating:** 2
**Confidence:** 4

**Summary:**

This paper proposes CC-Time, a framework for time series forecasting that attempts to leverage Pre-trained Language Models (PLMs). The stated contributions are: 1) Cross-Modality learning, using time series data and corresponding textual descriptions (semantic and statistical) to model channel correlations; and 2) Cross-Model fusion, via a CMF Block, to integrate features from the PLM branch and a dedicated Time Series (TS) branch.

**Strengths:**

1. The paper attempts a novel cross-modal fusion approach, and the idea of using PLMs to process channel correlations is explored. The design of the Cross-Model Fusion (CMF) Block is architecturally complex, utilizing multiple attention mechanisms to integrate information from the two heterogeneous branches.

2. Given the model's complexity, the authors have conducted numerous ablation studies. The experiments in Figure 3, Figure 9, and Appendix E attempt to demonstrate the necessity of the model's key components (though, as noted in the weaknesses, key ambiguities remain).

3. The model reports SOTA or competitive results on multiple datasets. The inclusion of comparisons against time series Foundation Models in a few-shot setting (Section 4.2 and Table 11) is a relevant experimental point, but the validity of these results is questionable given the methodological flaws.

**Weaknesses:**

1. **(Most Serious Issue) Dependency on Text Source and Generalizability**: The paper's primary methodological flaw is its dependency on an external LLM. While using an LLM to auto-generate text (Appendix A) solves the problem of missing text modalities in existing datasets, it builds a part of the model's performance on an uncontrolled, external black-box tool.

   * **Concerns about Text Quality**: We are concerned about the quality of the LLM-generated text.
   * **Concerns about Specific Datasets**: For a dataset like **Traffic** (with 862 channels), where channels represent sensors at different locations, can an LLM generate meaningful and **differentiated** descriptions for all 862 sensors? If the LLM just produces 862 copies of "This is a traffic sensor," the cross-modal innovation becomes meaningless.
   * **Question**: Could the authors provide examples of the (LLM-generated) text descriptions for several different channels from the **Traffic** dataset in their rebuttal?
   * **Motivation**: Overall, the motivation behind this article is somewhat strange. It seems that the text was added simply to introduce multimodality, and the quality of such text information is questionable.


2. **Source of Performance Gain: Semantics vs. Statistics**: Closely related to point 1, it is a reasonable inference that the performance gains may come more from the **"Statistical Information"** in the text description, rather than the "Semantic Description".

   * For datasets like Traffic, if (as we suspect) the "semantic descriptions" for different channels are highly similar or generic, the primary source of information for the PLM branch to differentiate channels becomes the "statistical information".
   * If this is the case, the novelty of this paper would be severely diminished, reducing the contribution to "a method for fusing statistical priors as auxiliary features into a time series model," rather than cross-modal semantic fusion.

3. **Ambiguous Ablation for Weakness #2**: **Appendix F.3 (Figure 9)** is critical for clarifying point 2, but its description is ambiguous.

   * **Question**: Can the authors explicitly confirm whether the "w/o Text" condition refers to: (A) removing *both* the semantic description and the statistical information, or (B) removing *only* the semantic description but *retaining* the statistical information?
   * This ambiguity also applies to the other experiments in that figure ("Add Noise", "Random Text"): are these interventions applied only to the semantic portion, or to both? Clarifying this is essential to evaluate the true contribution of "semantics".

4. **Necessity of the PLM Auxiliary Loss**: The model only uses the output of the TS branch during inference (Section 3.4), meaning the PLM loss ($\\mathcal{L}\_{plm}$) primarily functions as an auxiliary training objective.

   * \*\*Appendix F.2 (Figure 8)\*\*tests the loss weight $\\lambda$ and finds $\\lambda=0.6$ to be optimal, which does suggest the loss is useful.
   * **Question**: However, to clearly prove its "necessity," the most critical ablation would be the result for $\\lambda = 0$ (i.e., completely removing this loss). Can the authors provide the experimental data for $\\lambda = 0$?

5. **Weak Conclusion from CKA Analysis**: The CKA analysis in **Figure 5** is interesting, but the conclusion drawn is unconvincing.

   * The authors find that the CKA value of CC-Time is intermediate between the two model classes (PLM-based and TS-specific) and therefrom infer that the model captures "appropriate" complex features.
   * An intermediate CKA value is somewhat "inevitable" as the model is, by design, a hybrid (or average) of the two feature types. This result is not surprising.
   * **Question**: The authors fail to provide a convincing analysis of the causal link between an intermediate CKA value and low error, beyond simple correlation. Why is being "in the middle" necessarily "appropriate" or "superior"? The current analysis reads more as a phenomenon-observation rather than a deep insight.

6. **Misleading Terminology ("Cross-Model")**: The term "Cross-Model" is potentially misleading. In the community, this often implies the integration or interaction of multiple independent models (e.g., multiple expert PLMs). However, the paper's architecture involves only *one* PLM and *one* TS-specific model. Therefore, the fusion is more accurately a "Cross-Paradigm" (PLM vs. TS) fusion, not a "Cross-Model" fusion in the plural sense. The term inflates the architectural complexity.

**Questions:**

See the Weaknesses.

---

> ### Author Response · Authors · 2025-11-21
> **Response to Reviewer L1sA (Part I）**
>
> We would like to sincerely thank Reviewer L1sA for providing a detailed review regarding the channel text of CC-Time, auxiliary loss, and the CKA analysis. We have revised our paper accordingly.
>
> **W1:** Dependency on Text Source and Generalizability
>
> We want to clarify that CC-Time is designed from the perspective of leveraging the potential of Pre-trained Language Models (PLMs), rather than being a multimodal model strictly dependent on high-quality external text. We address specific concerns as follows:
>
> **Uncontrolled Black-box of LLMs：**
> - **Dataset-level Text Generation:** We clarify that channel semantic descriptions generation is performed on a dataset-level rather than for individual time series samples. Consequently, the volume of generated text is limited and finite, which allows for comprehensive manual verification to ensure reliability.
>
> - **Contrained Generation:** As detailed $\underline{\text{in Appendix A}}$, we employ structured prompts to constrain the LLM's output, ensuring it focuses on channel semantic information rather than open-ended generation.
>
> **Channel text quality:**
> To ensure the reliability of the generated text, we visualize the generated channel semantic text and assess the model's robustness of text quality.
> - **Visualization of Channel Semantic Text:** $\underline{\text{Figure 6 of the Appendix}}$ shows that, by leveraging the extensive knowledge in LLMs along with our prompts, the LLMs can generate accurate channel semantic information, providing a detailed explanation for each channel.
> - **Roubutness Analysis:** In the original paper, we have conducted channel text quality experiments $\underline{\text{ in Appendix F.3}}$. As shown in $\underline{\text{Figure 9}}$, adding noise to the channel text leads to only a slight drop in prediction accuracy, indicating that CC-Time is robust to minor degradation in the entire channel text quality.
>
>
> **Specific Datasets (Traffic):** Since the Traffic dataset does not provide specific channel names, we use only channel statistical information as the channel text for this dataset. For other datasets, such as Weather and CzeLan, we generate both channel semantic descriptions and statistical information to form the complete channel text.
>
> As shown in $\underline{\text{Table 1 of the paper}} $, CC-Time achieves good forecasting performance on Traffic using only channel statistical information and raw time series. This proves that our method generalizes well even without channel semantic information, effectively addressing the reviewer's concern about serious dependency on "meaningful" semantic descriptions.
>
>
> **Motivations:** We would like to clarify that CC-Time is designed from the perspective of effectively leveraging pre-trained language models (PLMs) for time series forecasting, rather than simply focusing on multimodal input.
> - **Core Contributions:** The core innovation of CC-Time is the PLM Branch and the Cross-Modal Fusion Block. The former effectively leverages the capabilities of PLM to model complex temporal dependencies and channel correlations, while the latter utilizes the strengths of both PLMs and time-series-specific models to enhance time series forecasting. Therefore, the main innovation of the model lies in its overall framework, rather than only in channel text generation.
> - **Regarding Text:** The text input in CC-Time serves as a prompt to activate the PLM's knowledge, which can be optional. Using only channel statistical text enables CC-Time to achieve superior performance. Furthermore, when the channel name is available, incorporating channel semantic descriptions can further enhance prediction accuracy.

---

> ### Author Response · Authors · 2025-11-21
> **Response to Reviewer L1sA (Part II)**
>
> **W2:** Semantics vs. Statistics: Performance Source
>
>
> We would like to clarify that the performance gain comes from semantic descriptions and statistical information. To demonstrate this, $\underline{\text{in the Appendix F.5 of the revised paper}}$, we conduct fine-grained text ablation study on ETTm1 and CzeLan, which possess rich channel semantic information and statistical information. We compare three settings:
> (a) removing both the semantic descriptions and statistical information (w/o Text)
> (b) removing only the semantic descriptions while retaining the statistical information (w/o semantic text)
> (c) removing only the statistical information while retaining the semantic descriptions (w/o statistical text)
>
> As shown in the following table, removing either the semantic descriptions or the statistical information leads to a drop in prediction accuracy. This confirms that both components make contributions to the forecasting performance, not just from the statistical information.
>
> Overall, the text input in CC-Time is optional. Semantic descriptions further enhance prediction accuracy, but even without them, when using only statistical text, the model still achieves superior performance compared with SOTA baselines. Importantly, our core contribution lies in effectively adapting the PLMs to unlock its potential, alongside the Cross-Modal Fusion Block that integrates the respective strengths of PLMs and time series models, rather than only in the use of channel text.
>
>
> | Models | CC-Time  | w/o Text | w/o semantic text| w/o statistical text|
> | --- | --- | --- | --- | --- |
> | Metrics | MSE / MAE | MSE / MAE | MSE / MAE | MSE / MAE |
> | ETTm1 | 0.375 / 0.377 | 0.382 / 0.382 | 0.380 / 0.379 | 0.378 / 0.379 |
> | CzeLan | 0.240 / 0.261 | 0.263 / 0.277 | 0.252 / 0.270 | 0.248 / 0.266 |
>
>
> **W3:** Clarification and ablation for the semantic descriptions and the statistical information
>
> We appreciate the reviewer for pointing out the ambiguity. We provide the clarification and additional detailed ablation studies below:
>
> **Clarification of the Original Figure 9 Settings:**
> - "W/O text" means removing both semantic descriptions and statistical information.
> - Both "Add Noise" and "Random Text" operate on the entire channel text, including both semantic descriptions and statistical information.
> We also revised $\underline{\text{the Appendix F.3 and Figure 9 of the revised paper}}$ to clearly state these settings.
>
> **Additional Ablation Study:**
> Referring to the $\underline{\text{Response to W2}}$ and $\underline{\text{Appendix F.5 of the revised paper}}$, our ablation analysis confirms that semantic descriptions and statistical text make contributions to forecasting performance.
>
>
>
>
> To more precisely assess the individual contributions of semantic and statistical information, we further inject light noise or random text separately into each part.
>
> As shown in the table, CC-Time exhibits robustness to perturbations in both parts. When either the semantic description or the statistical information is replaced with random text, the prediction performance drops significantly, indicating that both parts of the information are important for prediction.
>
>
> |  | CC-Time  | Add Noise of semantic text | Random Semantic text| Add Noise of Statistical text| Random  Statistical text |
> | --- | --- | --- | --- | --- | --- |
> | Metrics | MSE / MAE | MSE / MAE | MSE / MAE | MSE / MAE | MSE / MAE |
> | ETTm1 | 0.375 / 0.377 | 0.377 / 0.378 | 0.387 / 0.395 | 0.378 / 0.379 | 0.384 / 0.391 |
> | CzeLan | 0.240 / 0.261 | 0.243 / 0.265 | 0.258 / 0.273 | 0.244 / 0.263 | 0.252 / 0.272 |

---

> ### Author Response · Authors · 2025-11-21
> **Response to Reviewer L1sA (III)**
>
> **W4:** The necessity of the PLM Auxiliary Loss
>
> To assess the necessity of the PLM Auxiliary Loss, we include the case of λ=0 in $\underline{\text{Appendix F.2 of the revised paper}}$. The experiment is conducted on ETTm1 with a prediction length of 96.
> As shown in the table, the model performs the worst when λ=0, indicating that the PLM Auxiliary Loss is essential for effectively updating the PLM branch and enabling it to better capture time series patterns.
>
> |  $\lambda = 0$ | $\lambda = 0$.2 | $\lambda = 0$.4 | $\lambda = 0$.6 | $\lambda = 0$.8 |
> | --- | --- | --- | --- | --- |
> | MSE / MAE | MSE / MAE | MSE / MAE  | MSE / MAE | MSE / MAE |
> | 0.317 / 0.338 | 0.315 / 0.337  | 0.311 / 0.335 | 0.309 / 0.333 | 0.313 / 0.336 |
>
>
> **W5:** The CKA analysis
>
> Thank you for pointing out the issue. We acknowledge that the CKA analysis primarily serves as a phenomenon-level observation of our design. Our intention is to provide a perspective for cross-model comparison: time-series–specific methods exhibit high CKA values, PLM-based methods show low CKA values, and CC-Time falls in between. Our goal is not to claim that an intermediate CKA value necessarily leads to better forecasting performance, but rather to show that, from the CKA perspective, CC-Time integrates characteristics from both types of models.
>
> We also revise $\underline{\text{the CKA analysis of Section 4.4}}$.
>
> **W6:** Misleading Terminology
>
> Thank you for your suggestion. We agree that “Cross-Paradigm” more accurately reflects the fusion between the PLMs and the time-series–specific model. Due to the extensive appearance of this term across figures and text, we plan to systematically update this terminology in the final version to ensure thorough consistency.

---

### Author Response · Authors · 2025-12-03
**Summary of the Rebuttal**

Dear AC, SAC, PC,

We are sorry to hear about the recent OpenReview bug issue, and we fully support the proposed remedy actions.

We sincerely thank the new AC, the original AC, the PC, SAC, and all reviewers for your time and effort in coordinating and evaluating our submission. The feedback from each reviewer has been invaluable throughout the entire process. To facilitate the decision-making process,  we believe it is necessary to record the changes in our scores and summarize the Reviewers' Opinions and Concerns (up to Nov. 29).

1. **Score changes after rebuttal**. During the rebuttal, Reviewer NQT5 explicitly stated that their main concerns were addressed and raised the score. Reviewers L1sA, xBLS, and QNfh did not provide further responses.


**Tabel 1.** Initial score -> After Rebuttal.
| Reviewer | Intial Score | Rebuttal Score | Status |
| -------- | ------ | ------ | ------ |
| L1sA     | 2      | 2     | No response |
| xBLS     | 6      | 6            | No response |
| NQT5     | **4**      | **6**            | **Concerns addressed, Score raised** |
| QNfh     | 8      | 8            | No response |
| **Avg**  | **5**      | **5.5**            | |


2. **Reviewers' Opinions.**  Overall, the reviewers express positive evaluations of our work. Specific summary of opinions is as follows:

**Table 2.** Summary of Reviewers' Opinions.

| Opinions | Reviewer |
|--------| ------ |
|**Novel and well-motivated fusion framework**: Adaptive fusion, Complementary advantages from PLMs and time-series-specific models. | xBLS, NQT5, QNfh |
|**Innovative Correlation Modeling**: Multimodal input, Unlocking PLM's potential.  | xBLS, NQT5, QNfh |
|**Extensive Experiments**: SOTA results in full/few-shot settings, Extensive analysis. | L1sA, xBLS, NQT5, QNfh|
|**Clearly written**: Clearly written, Well-organized, Informative figures. | QNfh |


3. **Reviewer's Concerns.** In response to the reviewers’ valuable and constructive concerns, we have carefully revised our paper and conducted additional experiments accordingly. Here is the summary of the main concerns and our responses.

**Table 3.** Summary of Reviewers' Concerns.

| Concerns | Revisions | Reviewer |
|--------| ------ | ------ |
| **Channel text quality** | We clarify that channel text quality is ensured through structured prompt design and comprehensive manual verification, and is further validated by the robustness analysis in Appendix F.3. | L1sA, QNfh |
| **The CKA analysis** | We clarify that the CKA analysis (Figure 5) serves as a phenomenon-level observation to justify our design. | L1sA |
| **Ablation study** | We add ablation studies: 1) comparing semantic descriptions with statistical text (Appendix F.5) and 2) testing the removal or replacement of PLMs (Appendix F.1), which confirm the effectiveness of these components.  | L1sA, NQT5|
| **Discussion with multimodal time series models** | We discuss the differences between CC-Time and existing methods from various perspectives (Appendix G), which further highlight the unique advantages of CC-Time. | xBLS|
| **Efficiency analysis** | We analyze the computational complexity of each module and demonstrate our efficiency superiority in Appendix F.6. | xBLS, QNfh |
| **Correlation Visualization** | Visualization analysis (Appendix H) on the PLMs and extractor that effectively model local and global correlations respectively. | NQT5|

We once again sincerely appreciate your time and the additional effort required to evaluate our paper. We hope that this summary assists the new AC in reducing their workload when making the final decision.


Best regards,

Authors

---

### Meta-Review · Area_Chair_E7mD · 2026-01-13

**Summary:**

This paper proposes CC-Time, a dual-branch forecasting framework that combines a PLM branch (using channel descriptions) with a time-series branch, connected via a cross-modality learning scheme and a cross-model fusion block. Extensive ablations and empirical results in both full-data and few-shot settings have shown that multimodal fusion can help model channel correlations. However, The main concerns are around clarity and validity of the proposed method. In particular, the core methodology is undermined by a heavy dependence on externally LLM-generated text descriptions, which are uncontrolled, potentially low-quality at scale (e.g., hundreds of channels), and make the claimed gains difficult to attribute or reproduce. Also, the paper does not cleanly justify whether improvements come from semantic descriptions versus injected statistical information, and several key ablations are missing/ambiguous. Although some ablation are provided in the rebuttal stage, the results are not convincing as the performance difference over ETTm1 is quite minor. None case studies are provided to explain when and where the semantic information/statistical information would be helpful. Therefore, this paper cannot be accepted with the current status.

**Reviewer Concerns:**

One main concern is that the performance may depend on an uncontrolled black-box LLM and unclear whether generated descriptions are meaningful at scale, e.g., 862 traffic channels. While an ablation study were provided, the authors do not provide representative examples of generated text for many channels and assess the assess sensitivity when text is degraded.

Another main concern is that the improvements may come primarily from embedding statistical summaries rather than true “semantic” modality. An ablation have been provided in the rebuttal stage, however, the results are not convincing as the performance difference over ETTm1 is quite minor.

**Reviewer Scores:**

Reviewer NQT5 indicates that he will increase the score from 4.

---

### Decision · Program_Chairs · 2026-01-26

Reject